# Bias in CMIP6 models as compared to observed regional dimming and brightening

Kine Onsum Moseid[1], Michael Schulz[1,2], Trude Storelvmo[2], Ingeborg Rian Julsrud[2,1], Dirk Olivié[1], Pierre Nabat[3], Martin Wild[4], Jason N. S. Cole[5], Toshihiko Takemura[6], Naga Oshima[7], Susanne E. Bauer[8], and Guillaume Gastineau[9]

[1]Norwegian Meteorological Institute, Research Department, Oslo, Norway
[2]University of Oslo,Department of Geosciences, Section for Meteorology and Oceanography, Oslo, Norway
[3]Centre National de Recherches Meteorologiques (CNRM), Universite de Toulouse, Météo-France, CNRS, Toulouse, France
[4]Institute for Atmospheric and Climate Science, Swiss Federal Institute of Technology (ETH), Zurich, Switzerland
[5]Canadian Centre for Climate Modelling and Analysis, Environment Canada, Victoria, British Columbia, Canada
[6]Climate Change Science Section, Research Institute for Applied Mechanics, Kyushu University, Fukuoka, Japan
[7]Meteorological Research Institute, Japan Meteorological Agency, Tsukuba, Ibaraki, Japan
[8]Center for Climate Systems Research, Columbia University, NASA Goddard Institute for Space Studies, New York, NY, USA
[9]LOCEAN, IPSL, Sorbonne Université, IRD, MNHN, CNRS, Paris, France

**Correspondence:** Kine Onsum Moseid (kristineom@met.no)

**Abstract.** Anthropogenic aerosol emissions have increased considerably over the last century, but climate effects and quantification of the emissions are highly uncertain as one goes back in time. This uncertainty is partly due to a lack of observations in the pre-satellite era, making the observations we do have before 1990 additionally valuable. Aerosols suspended in the atmosphere scatter and absorb incoming solar radiation, and thereby alter the Earth's surface energy balance. Previous studies show that Earth system models (ESMs) do not adequately represent surface energy fluxes over the historical era. We investigated global and regional aerosol effects over the time period 1961-2014 by looking at surface downwelling shortwave radiation (SDSR). We used observations from ground stations as well as multiple experiments from eight ESMs participating in the Coupled Model Intercomparison Project Version 6 (CMIP6). Our results show that this subset of models reproduces the observed transient SDSR well in Europe, but poorly in China. We suggest that this may be attributed to missing emissions of sulfur dioxide in China, sulfur dioxide being a precursor to sulfate, which is a highly reflective aerosol and responsible for more reflective clouds. The emissions of sulfur dioxide used in the models do not show a temporal pattern that could explain observed SDSR evolution over China. The results from various aerosol emission perturbation experiments from DAMIP, RFMIP and AerChemMIP show that only simulations containing anthropogenic aerosol emissions show dimming, even if the dimming is underestimated. Simulated clear sky and all sky SDSR do not differ largely, suggesting that cloud cover changes are not a dominant cause to the biased SDSR evolution in the simulations. Therefore we suggest that the discrepancy between modeled and observed SDSR evolution is partly caused by erroneous aerosol and aerosol precursor emission inventories. This is an important finding as it may help interpreting whether ESMs reproduce the historical climate evolution for the right or wrong reason.

# 1 Introduction

Aerosol particles scatter and absorb radiation and change the radiative properties of clouds, thereby altering Earth's energy balance (Boucher et al., 2013). Anthropogenic aerosol emissions have substantially increased over the last century, but the quantification of the effect has been characterized by large uncertainties. Earth system models (ESMs) are evaluated based on their ability to reproduce the climate evolution of the past 165 years, and the sparsity of aerosol-related observations in the pre-satellite era plays a dominant role in the uncertainty connected to these historical experiments. An improved understanding of the historical aerosol effect would increase the accuracy and credibility of ESMs future climate projections.

Aerosol particles cause changes in the amount of sunlight reaching the surface together with changes in insolation, cloud cover, water vapor and other radiatively active gases (Wild et al., 2018). Extraterrestrial influences like the 11-year cycle of the sun have not created any important trends on decadal time scales in Earths surface solar radiation in the past century (Eddy et al., 1982; Wild, 2009). Water vapor amount has not changed sufficiently in recent decades to have an effect on decadal fluctuations of incoming sunlight at the surface (Wild (2009), Wang and Yang (2014), Yang et al. (2019), Hoyt and Schatten (1993), Ramanathan and Vogelmann (1997), Solomon et al. (2010)), and radiatively active gases dominate in the longwave spectrum (Ramanathan et al. (1989)).

The relative roles of clouds, aerosols and their interactions in historical variations of surface downwelling shortwave radiation (SDSR) are still disputed, but previous studies have found that aerosol effects dominate on multidecadal timescales, while cloud effects are relevant on shorter timescales (Wild (2016), Romanou et al. (2007)). Aerosol effects can be divided into the direct and indirect effect. The direct effect is the scatter or absorption directly caused by a dry aerosol, also called the aerosol-radiation-interaction (ari) (Boucher et al., 2013), and the indirect effect is how aerosols change properties in clouds, also called aerosol-cloud-interactions (aci). Aci includes both a change in cloud lifetime and most importantly a change in cloud albedo, making the cloud appear brighter (Boucher et al., 2013).

Assuming aerosol effects dominate the multidecadal timescales, SDSR can serve as a proxy for aerosol effects. The Global Energy Balance Archive (GEBA) dataset contains measurements of SDSR as far back as in 1922 (Wild et al., 2017), and as such represents a unique and valuable dataset for evaluation of simulated aerosol effects prior to the satellite era.

Observed SDSR from the GEBA dataset reveals a widespread negative trend from the 1950s to the late 1980s, commonly referred to as "global dimming" (Liepert (2002), Wild (2016)). The magnitude of this dimming differs vastly between regions, which is expected if the cause of dimming were regionally different increases in aerosol emissions, as has been proposed by Wild et al. (2007), Sanchez-Romero et al. (2014), and Wild (2016). In some areas a positive trend in SDSR follows the dimming, and this SDSR increase has been termed "brightening" (Wild et al., 2005). Brightening is connected to the reduction in anthropogenic aerosol emission (Nabat et al., 2014). Fewer particles suspended in the air allow for more sunlight to reach the surface and thus an increase in the measured SDSR. Previous studies show that historical simulations from ESMs do not reproduce the observed global transient development of SDSR (Storelvmo et al. (2018), Wild (2009), Allen et al. (2013), Wild and Schmucki (2011)). The cause of this discrepancy is not known, but may be connected to uncertainties in aerosol emission

inventories of the past, or, as Storelvmo et al. (2018) suggested, other uncertainties concern how models treat processes that translate aerosol emissions into radiative forcing.


In this study we use gap-filled data based on the GEBA dataset, together with several recent CMIP6 historical model experiments from eight climate models to investigate the aerosol effect in the time period 1961-2014, globally and regionally. In the middle of this time period (around the late 1990s), the main region of high anthropogenic aerosol emissions shifted from Europe and North-America to Asia. We have chosen to focus on the regions of Europe and Asia in this study, as the models

exhibit diverging abilities to reproduce the observed SDSR in these regions. We also use observational cloud cover data to briefly assess the role of cloud cover in the historical development of SDSR. We explore the relation between regional SDSR and aerosol emissions using a set of ESM experiments with differing aerosol emissions; some have pre-industrial aerosol emissions, while others use the most recent and best available historical aerosol emission inventory (Hoesly et al., 2018). This paper thereby provides new insights into the question of whether state-of-the-art ESMs can adequately reproduce a part of

the changes in the surface energy budget over the historical era. This is in turn an important indication of whether the ESMs reproduce the dominant processes governing the historical climate evolution.

The paper is structured as follows: In Section 2 we begin by presenting the two observational datasets used, followed by a detailed description of the experiments simulated by the eight models chosen to be part of this study. The methods used to

obtain and analyse the data finalize Section 2. The results are presented in Section 3, starting with a global view of dimming and brightening before focusing on regional assessments of SDSR, clear sky SDSR, and cloud cover. Section 4 discusses the implications of our results and how they compare to previous studies, before final conclusions are presented in Section 5.

## 2   Data and Methods

### 2.1   Observations

The Global Energy Balance Archive (GEBA) holds data from ground-based stations measuring energy fluxes at the Earth's surface around the globe (Wild et al., 2017). Pyranometers were used in most of the measurement sites, which have an accuracy limitation of 3-5 % of the full signal (Michalsky et al. (1999), Wild et al. (2013)). We use the monthly mean data from 1487 stations in the time period 1961-2014 measuring downwelling shortwave radiation. The GEBA data set has been complemented

by a machine learning technique (random forests (Breiman, 2001)) as explained in Storelvmo et al. (2018) to cover time periods of missing observations in the measurements and facilitate comparison to the gridded model data. This allows for all 1487 stations to have data on each time step, so that all regions have a complete record and the same amount of stations throughout the entire time period in question.

Monthly mean cloud cover data is provided by the Climatic Research Unit (University of East Anglia) and NCAS, and we are

using version 4.02 of this dataset (CRU). CRU covers the period 1901-2017 (Harris et al., 2020) and consists of a climatology

made from measurements at meteorological stations around the globe, interpolated to a 0.5$^o$ latitude/longitude resolution grid covering continental areas. Information on interpolation methods and procedures used to create the gridded data set are given in Harris et al. (2020) and references therein. In short, CRU has its foundation in station data, but is interpolated to a grid using angular-distance weighting. The cloud cover variable is largely derived as a secondary variable, based on measurements of other parameters such as sunshine hours and diurnal temperature range.

## 2.2 Models and CMIP6

Eight climate models (NorESM2, CanESM5, MIROC6, CESM2, CNRM-ESM2-1, GISS-E2-1-G, IPSL-CM6A-LR, MRI-ESM2-0) were chosen for this study, based on available data and their involvement in relevant model intercomparison projects within the Coupled Model Intercomparison Project Phase 6 (CMIP6) (Eyring et al., 2016). As this study focuses on dimming and brightening, we have chosen experiments from model intercomparison projects (MIPs) that include perturbed historical simulations with which one can single out the effect of anthropogenic aerosol emissions in our diagnostic variables. An overview of models and experiments can be found in Table 1. This section will give a more detailed description of the experiments in Table 1 and explain why they were chosen.

Every model that takes part in CMIP6 has to deliver a set of common experiments, among these is the *historical* simulation. As can be seen in Table 1 all the models have provided historical simulation results. All other experiments listed in Table 1 are simulations covering the historical period (1850-2014) but with specific alterations dependent on what model intercomparison project they are a part of.

The Detection and Attribution Model Intercomparison Project (DAMIP) has the goal of improving estimations of the climate response to individual forcings (Gillett et al., 2016) and includes three relevant experiments. One experiment traces exclusively the impact of anthropogenically emitted aerosols as forcing agents over the historical period, and is called *hist-aer*. This means no anthropogenic greenhouse gas emissions or natural climate forcings are used in this simulation. The *hist-nat* experiment consists of only the perturbations due to the evolution of the natural forcing, e.g. from stratospheric aerosols of volcanic origin and solar irradiance variations. Finally, the *hist-GHG* experiment has only forcings from changes in the well mixed greenhouse gases. These experiments were chosen as they give a unique insight into how a fully coupled climate model attributes responses over the historical period to the different climate forcings.

While DAMIP provides a good framework for one of the main questions in CMIP6, namely how the Earth system responds to forcing, RFMIP, the Radiative Forcing Model Intercomparison Project, focuses on understanding the forcing itself. RFMIP contains a large set of experiments to further understand the radiative forcing of the past and the present (Pincus et al., 2016). We use two experiments from RFMIP, both with sea surface temperatures prescribed to pre-industrial values. One experiment includes both anthropogenic and natural aerosol emissions (*piClim-histall*) while the other only includes anthropogenic aerosol emissions (*piClim-histaer*). When sea surface temperatures are kept to pre-industrial values, the global surface temperature development stalls, and the simulation will keep to first order a pre-industrial climate. Sea surface temperatures changes would

have an effect on cloud cover, which in turn can affect SDSR. These *piClim*-experiments will show the direct atmospheric forcing on SDSR due to greenhouse gases and aerosols, alone or in combination, without including cloud cover changes induced by global warming.

The third MIP included in this study is the Aerosol Chemistry Model Intercomparison Project (AerChemMIP), which is
designed to answer questions regarding the specific effect of aerosols and other near-term climate forcers (NTCF) on climate. NTCFs include methane, tropospheric ozone, aerosols and their precursors (Collins et al., 2017). Three experiments have been selected from AerChemMIP, *histSST*, with all forcing agents included, and two perturbations which have pre-industrial aerosols emissions: (*hist-piAer*) and (*hist-piNTCF*). The hist-piNTCF experiment has in addition pre-industrial NTCF levels for ozone. A difference in these two simulations would only appear if ozone concentrations were computed in an interactive chemistry
scheme. These two simulations are coupled and are comparable to the historical experiment. The experiment *histSST* uses all forcing agents and the sea surface temperatures derived from the historical simulation so that the temperature evolution, and hence its effect on SDSR, should be similar to the historical experiment, but removes responses involving a coupled ocean. These experiments together with the historical experiment were chosen to differentiate between historical changes in aerosol and tropospheric ozone, or whether a mixing layer in the ocean may have had an effect on dimming.
Data from all experiment ensembles from each of the MIPs listed above provide useful information on the role of anthropogenic aerosol emission in dimming and/or brightening.

## 2.3   Methods

The GEBA stations have been divided into regions based on the country and continent. The number of stations in a region is presented together with the first results in the caption of Figure 2. The number of stations per region remains constant
throughout the time period because of our gap filling approach. A figure with the spatial distributions and trend of SDSR per station in GEBA used in this study is found in Figure 1 in Storelvmo et al. (2018).

All model output and CRU results have been co-located to GEBA station locations using the nearest neighbour method. This entails that if two GEBA stations are within one grid box of a model, data from that grid box will be retrieved twice by nearest neighbour interpolation, as every station has been weighted equally. A global mean is defined here as the mean of a variable
across all GEBA station locations. A regional mean is a mean of a variable across the GEBA station locations registered to that same region in the GEBA data. When a result is shown as an anomaly, as opposed to an absolute value, the general formula has been to subtract the baseline value, defined as the mean of the first five years of the investigated time period (1961-2014), from the timeseries in question. To clarify - first an average value per year per region is calculated, and then a new mean is created from the first five years of this timeseries. This 5-year-mean is then subtracted from each year in the timeseries for the
region in question and presented as an anomaly. We will often present data as 6-year-averages, as yearly variabilities are not the focus of this study. These 6-year-averages are simply made by dividing the timeseries over 54 years(1961-2014) in nine equal intervals and average these intervals together. When the atmospheric burdens of $SO_4$ is shown together with observed

SDSR from GEBA the timeseries have been smoothed using a 10-year running mean, and this is the only data in the paper shown using this smoothing technique.

The "baseline" values for global SDSR and cloud cover in the models and observations of this study can be found in the Appendix in Table A1.

The model data has been retrieved from The Earth System Grid Federation (ESGF) (Cinquini et al., 2014). ESGF is a data management system consisting of multiple geographically distributed nodes that coordinate through a peer-to-peer (P2P) protocol (Fan et al., 2015). We have used three ensemble members for the *historical* experiment to present internal variability in

the models, and one ensemble member from the rest of experiments shown, as not every experiment had requested more than one simulation. Table A2 in the Appendix shows the resolutions, aerosol schemes, and aerosol complexity of the models in this study, and Section A2 explains the variables and variant labels downloaded.

## 3 Results

### 3.1 Model variability

Figure 1 shows SDSR anomaly for each model of the study co-located to all GEBA stations, 1487 in total as compared to the observed SDSR anomaly. The aerosol effective radiative forcing (Aerosol ERF) corresponding to each model is obtained from Smith et al. (2020) and is listed in each panel to illustrate the strength of the aerosol radiative effect in the model.

Each climate model has its own internal variability and thereby represents its separate climate systems. SDSR is a highly variable metric on a year-to-year basis, which can be seen both in the GEBA data in black in Figure 1 and in following a

single ensemble member per model. Within each model ensemble one can see that no member is equal to another, which is a clear signal of the internal variability of each model. The spread of all three ensemble members in a 6 year period can be read from the height (interquartile range) of the boxes in the 6-year-intervals, note that this spread is dominated by large interannual variabilities within each member. One example is GISS-E2-1-G, where each ensemble member has large interannual variabilities, the boxes present long whiskers and large interquartile ranges, but when comparing the ensemble member 6 year

means one by one they mostly agree on their magnitudes of SDSR anomaly, so the intra-ensemble-spread is not large for GISS-E2-1-G. We find (not shown here) that the model with the least interannual variabilities is CNRM-ES2-1, while the model with the largest inter-ensemble disagreements is CanESM5.

Figure 1 also shows that the models in general do not agree with the observed global SDSR anomaly, shown in black. Dimming and brightening are tendencies in surface radiation that are observed on longer than interannual timescales, with this in mind

SDSR from models will in general be presented as 6-year means for the remainder of this paper. The model MRI-ESM2-0 is showing the most similar SDSR evolution compared to the observed data according to Figure 1.

The model with the strongest aerosol ERF is CESM2, while the weakest aerosol ERF is presented by IPSL-CM6A-LR.

## 3.2 Dimming and brightening

The change in SDSR in the *historical* simulations from the eight models is presented together with GEBA data in Figure 2. Panel (a) of this figure corresponds to the results shown in Figure 1. Each model graph in Figure 2 represents the ensemble mean of the model in question averaged over 6 years, based on three ensemble members. GEBA data is shown in black, also as 6-year averages, but with the yearly time series shown in grey in the background. Model simulations show small changes of global SDSR compared to observations (Fig 2a). Global SDSR is observed to decrease over the 1487 stations until late 1980's before increasing again, clearly showing the global "dimming" and "brightening" as found in previous studies listed in the introduction.

None of the models outperform one another globally, and there is a discrepancy of about 2-3 $W/m^2$ between models and observations. To further identify from where this discrepancy originates, we consider some geographical regions separately. Asia and Europe are relevant regions in regards to anthropogenic aerosol emissions (as explained in Section 1) and thereby also relevant to global dimming and brightening. The historical SDSR evolution in Europe and Asia are presented in Figure 2 (b) and (c), respectively. European SDSR is relatively well represented by the model simulations. The yearly GEBA timeseries has values within the shaded area, that is showing the standard deviation of the total of 24 model ensemble values, in almost every 6 year period in Europe. The dimming in Europe is believed to have started before 1961 (Wild, 2009), which partly explains why the initial European dimming in Figure 2(b) is weak. GEBA shows a short-term positive anomaly between 1970 and 1980, which is not caught by the models. This peak is currently unexplained, but a short assessment of its possible association to changes in cloud cover is found in Section A1 in the Appendix.

There is generally a large discrepancy between model simulations and observations of SDSR in Asia, as seen in Figure 2(c). The ground stations in Asia show a noticeable trend change in SDSR in the transition from 1980s to 1990s that is not apparent in the model simulations. The historical model simulations show a consistent negative trend during the entire historical period in question in Asia. Historically, countries with relatively high emissions in Asia include India, Japan, and China (Hoesly et al., 2018), and the SDSR evolution for each of these countries are shown in Figure 2(d), (e), and (f), respectively.

Figure 2 (d) shows that the models capture a relatively strong negative trend of SDSR in India, with MIROC6 being the model with the most modest trend. There are evident differences between observations and simulations in both Japan and China. Ground stations in Japan show a sharp decrease in SDSR until the early 1970s followed by some variations until a new minimum value is reached around 1990 before an increase in SDSR is measured. The minimum value around 1990 and the following positive trend is similar to that of China. Japan is downwind of the Asian continent and thus believed to be influenced by aerosol emissions from China. Model simulations do not capture the magnitude of dimming in Japan, or the apparent brightening in the 1990s. The timing of minimum SDSR occurs differently in models, which was also seen in Figure 2(a).

Observations from China (Figure 2(f)) show a trend change in SDSR similar to the one identified in Figure 2(c) for Asia as a whole, with the minimum value found in 1989. We note that China consists of 119 GEBA stations while Asia as a whole consists of 311 stations, thus the Asian average is largely impacted by SDSR as measured in chinese stations. In general the

historical model simulations show dimming throughout the historical period in China, meaning none of them show a similar trend change to the one from the observational data set. This post-1990 trend change is a source of discussion within the field, and a thorough assessment, relevant to the conclusions from this study, is found in Section 4.1.

### 3.3 Dimming and brightening over China in various CMIP6 experiments

In order to understand which forcing agents are responsible for the overall trends in SDSR in the models, we now investigate China for the experiments listed in Table 1. Figure 3(a) shows perturbed historical simulations as performed in DAMIP together with observations of SDSR. DAMIP has two experiments without historical anthropogenic aerosol emission (dashed/*hist-nat* and stippled/*hist-GHG* lines), and one experiment with historical anthropogenic aerosol emissions (solid lines/*hist-aer*). The experiment *hist-aer* is the only experiment in DAMIP exhibiting a distinguishable dimming signal. SDSR from *hist-aer* shows patterns similar to the *historical* simulations with continuous dimming throughout the period, unlike the observed SDSR. SDSR in the experiments *hist-nat* and *hist-GHG* do not show signs of dimming or brightening over the investigated period in China, which confirms that water vapor or stratospheric aerosols are not the dominant cause for multidecadal dimming signals in the fully coupled historical model simulations. This is supported by previous work as mentioned in the introduction.

Out of the three experiments from AerChemMIP only *histSST* has prescribed sea surface temperatures and contains changes in anthropogenic aerosol emissions. This is consistent with the time evolution of SDSR in *histSST* as the simulations diverge from the other simulations as time progresses (Fig 3b). Keeping in mind that *histSST* also has anthropogenic GHG emissions in addition to natural forcers, the only difference from *histSST* to the *historical* experiment is the absence of a coupled ocean and the use of prescribed sea surface temperatures. The model MRI-ESM2-0 presents the strongest dimming in both DAMIP and AerChemMIP. The simulations with pre-industrial aerosols (*hist-piAer*) and pre-industial near term climate forcers, including aerosols and ozone (*hist-piNTCF*) show very small or negligible changes in the SDSR over the time period considered.

Recall that the experiments of RFMIP utilize pre-industial SST's, meaning essentially there is no global warming in these experiments. In the RFMIP experiments shown in Figure 3(c) both *piClim-histaer* and *piClim-histall* contain anthropogenic aerosol emissions, and all simulations show a continuous dimming throughout the period. There is no clear distinction between experiments containing GHG emissions in addition to anthropogenic aerosol emissions (solid lines/*piClim-histall*) and the experiments only containing anthropogenic aerosol emissions (stipled lines/*piClim-histaer*). This implies that greenhouse gases without their global warming effect do not affect multidecadal all sky SDSR in a significant way over China throughout the period, again supported by previous work mentioned in the introduction.

Overall there is a clear difference in SDSR between experiments that include anthropogenic aerosol emissions and experiments that do not. Dimming is apparent in every simulation containing anthropogenic aerosol emissions, but absent in the simulations using aerosols maintained at constant pre industrial levels. This points to anthropogenic aerosol emissions playing a key role

in dimming. Whether the sea surface temperature is pre-industrial, prescribed historical, or decided by a coupled ocean model seems to be unimportant for the SDSR temporal evolution in China in most models.

No distinct flattening or brightening is identified in any of the simulations in which dimming is identified, and therefore none of the model simulations show a temporal evolution of SDSR close to the one seen in observations over China.

All sky SDSR changes can be further decomposed by the models into a clear sky contribution as well as a contribution from changes in cloud cover or other cloud properties. In the next section we present the decomposed contributions to all sky SDSR in China to further understand the discrepancy seen in Figure 3.

### 3.4    Clear sky SDSR and cloud cover in China

So far we have only evaluated all sky SDSR, which is influenced by clouds and any aerosol radiative effects. Table 2 shows changes in cloud cover, all sky SDSR, and clear sky SDSR within three different time periods for the models and observational data sets of this study. Between the years 1961 and 1989 GEBA shows a strong negative change in all sky SDSR in Figure 2(f). In Table 2 we thus show changes in this time period by making two 3-year means and subtracting them from one another. This is done to avoid extreme values as we are working with metrics exhibiting large year to year variations. This has been done for

two additional time periods which have been chosen based on the temporal development in the all sky SDSR as measured by GEBA in China ( see Figure 2(f)), summarized in the second lowest row in Table 2.

In the first time period the models do not agree on the sign of cloud cover change, and the simulated all sky SDSR is weaker than the observed one, which was already established in the previous section. Clear sky SDSR do not differ largely from all sky

SDSR within the models. For some models the negative change in clear sky SDSR is stronger than in all sky SDSR, meaning that the aerosol direct effect may contribute significantly to dimming for these models. The aerosol indirect effect changes the radiative properties of clouds in two ways, by making them appear brighter, and by altering their lifetime (Boucher et al., 2013). Therefore, a weak change in cloud cover followed by a strong change in all sky and clear sky SDSR points to both the direct and the brightening indirect aerosol effect being the primary cause of SDSR change, as an altered lifetime of clouds

would imply cloud cover changes.

In the second time period GEBA shows a positive change (which will be further discussed in Section 4.1), and CRU shows a cloud cover change of +3.0 %. Intuitively, an increase in cloud cover would not create a brightening at surface level. The observations are thus not consistent in this time period if only cloud cover effects were important. The models disagree in their

sign of cloud cover changes, all sky, and clear sky SDSR. In the final time period where GEBA shows a weak slightly positive change in all sky SDSR, every model in this study shows a dimming. All models apart from MIROC6 show simulated clear sky SDSR changes that are stronger than the changes found in all sky SDSR. Together with the inconsistent simulated cloud cover and all sky SDSR changes for this time period we suggest that both direct and indirect aerosol effect are responsible for

the changes in SDSR found in the model simulations.


All models show dimming in clear sky and all sky SDSR in the first and last time period. Some models show a weak positive change in all sky SDSR in the same period as GEBA presents a strong brightening. Both observed and simulated changes in cloud cover neither act as a brightening mask for clear sky dimming nor are convincingly a cause of dimming/brightening in either observed or simulated all sky SDSR. A rough calculation of the effect of 1 % cloud cover increase on SDSR in China

is found in Section A3 in the Appendix, indicating that such an increase could result in a dimming of 1-3.5 Wm$^{-2}$. As such it shows that observed and modelled changes in cloud cover, as reported in Table 2, can lead to important contributions to the dimming and brightening signals in SDSR. However, this calculation is idealized, does not isolate the cloud cover change effect in the model results and does not explain the inconclusive data reported in Table 2. It is important to note that the robustness of observed cloud cover changes must be verified by satellite observations, which goes beyond the scope of this study.


In section 3.3 we showed that dimming was only apparent in simulations that included anthropogenic aerosol emissions. In this section we found the clear-sky SDSR to be close in value or even stronger than all-sky SDSR, indicating the simulated dimming is primarily caused by aerosol effects. Table 2 underlines previous findings, dimming in models are overall weaker than in observations. The next section will then show how the simulated aerosol burdens are connected to SDSR.

## 3.5 Atmospheric burden of SO$_4$

In the atmosphere, the *presence* of a reflective aerosol is the cause for scattered shortwave radiation, and the emission of its precursor is only an indirect indicator of its presence. All CMIP6 simulations mentioned above have utilized the same anthropogenic sulfur dioxide gas emissions, however the resultant dimming differed considerably. SO$_4$ aerosol burdens should be more closely linked to the radiative effect. Therefore, we present here also the simulated anomalies in burden of SO$_4$ in

the various models over Europe, a location where dimming and brightening is fairly well represented in simulations, and over China, where dimming and brightening is poorly represented in simulations (Figure 4(a) and (b) respectively). The sulfate burdens are co-located to GEBA station locations in the respective regions. As expected, sulfate aerosols have an important role in European dimming and brightening, as the simulated burdens of SO$_4$ show a strikingly similar pattern (but with opposite sign) as the observed SDSR over Europe for all models. The maximum burdens are found in the early to mid 1980s depending

on the model, and the minimum SDSR around the same time. The various models differ in the magnitude of change in SO$_4$ burden over Europe but all show similar tendencies. MRI-ESM2-0 is the model with the largest changes, and GISS-E2-1-G is the model with the smallest changes in SO$_4$ burden. The same is observed over China, where MRI-ESM2-0 has an SO$_4$ burden at the end of the time period which is more than double the burden of the other models (except NorESM2). In contrast to Europe, the observed SDSR evolution does not mirror well the simulated SO$_4$ burden timeseries over the GEBA stations

in China. In order for the SO$_4$ burden to be the main cause of the observed changes in SDSR, the Asian SO$_4$ burden would have to peak around the late 1980s, which is not seen in the models in Figure 4(b). All the simulated historical SO$_4$ burdens increase until 2010, showing no signs of either a trend change or a flattening of aerosol induced dimming. Assuming GEBA

data provide a reasonable representation - within uncertainty bounds as discussed in section 4.1 - of the historical development of SDSR and implicitly sulfur burdens in China, the problem in $SO_4$ burden must come from either the emissions, aerosol

formation, transport or the removal processes of $SO_4$.

It appears, however, that the simulated burdens of $SO_4$ co-located to GEBA stations in China follow quite closely the time series of emitted $SO_2$ in the climate models over China (shown in Appendix Figure A2), which indicates that $SO_4$ formation and export of sulfur from the Chinese region remains rather similar in the period investigated. Following the logic that emis-

sion correlate with burden which again anti correlates with SDSR changes, the temporal development of SDSR seen in GEBA cannot be explained from the current emission inventories, given sulfate aerosols play an important role in SDSR in China.

## 4    Discussion

The climate effect of aerosol emissions over the industrial era is poorly constrained, in part due to lack of observations and

uncertainty in emissions. The uncertainty in past aerosol climate effects is an important reason for the large spread in climate projections for the future. Here, we investigate the effect of aerosols in GEBA which provides valuable observations of historical shortwave radiation at the surface.

We have shown that a subset of models participating in CMIP6 do not accurately represent the observed dimming and brighten-

ing trends globally and regionally in their *historical* simulation. This is comparable to that of Storelvmo et al. (2018) and Wild and Schmucki (2011), who showed that the CMIP5 and CMIP3 ensemble mean SDSR globally co-located to GEBA stations does not represent dimming or brightening. Our findings show that reproducibility of SDSR has not improved from CMIP5 to CMIP6. We find that most models show an underestimation of changes in SDSR to observations, and the development over time greatly differs between model and observations, especially in China. This is in agreement with Allen et al. (2013) who

studied the CMIP5 ensemble mean and found a continuous dimming trend over China, but with a severely underestimated magnitude of modelled clear-sky SDSR during the dimming period compared to a clear-sky proxy based on GEBA data.
The simulated SDSR on decadal timescales over China does not differ significantly when comparing the RFMIP experiments (Figure 3) to the historical experiment. RFMIP experiments have pre industrial sea surface temperatures, and thus do not include global warming induced cloud cover changes. When experiments with historical cloud cover changes show dimming in

the same magnitude as experiments without historical cloud cover changes, the dimming can be assumed to be dominated by aerosol effects in China. This complements the findings by Folini and Wild (2015) where sea surface temperatures correlate to cloud cover, not aerosol effects. Table 2 showed inconclusive connections between modelled and observed cloud cover, but clear connections between clear sky SDSR and all sky SDSR, again pointing to aerosol effects dominating SDSR time evolution in China.


The climate models strongly underestimate the dimming observed in China, in addition to not representing the post-1990 trend change. This trend change is the topic of discussion in the next section.

## 4.1 The post-1990 trend change in China

Several studies have tried to explain the trend change as presented here by GEBA in China in the transition from the 1980s to the 1990s. Streets et al. (2006) proposed a peak in combined emissions of $SO_2$ and black carbon in 1988-1989 as a possible explanation. A later study questions the quality of the observational data showing the trend change (Tang et al., 2011), while recent studies propose the post-1990 initial, strong brightening is to a considerable extent an artifact of a nation wide change in SDSR measurements (Wang and Wild (2016) Yang et al. (2018)). The change in SDSR measurements include a replacement of SDSR instrumentation, an increase in measurement frequency and in addition an update in the classification of SDSR stations, and Wang and Wild (2016) conclude that the upward trend ("jump" between 1990-1999) should be considerably weaker, and that only 20 % of the "jump" has actual physical causes. Yang et al. (2018) homogenized the data from Wang and Wild (2016) and Wang et al. (2012) and presented a new SDSR evolution (results can be seen in Yang et al. (2018) Figure 10). The newly homogenized data exhibit a significant dimming trend (-6.13 +- 0.47 W/m$^2$/decade) between 1958-1990, a flattening of the curve in 1991-2005, followed by a brightening trend (6.13 +- 1.77 W/m$^2$/decade) between the years 2005-2016. We can use Figure 2(f) to compare our model data to these homogenized data, and see that even without a larger "jump" in the data around 1990 there are still large discrepancies between model and observation, both in the form and magnitude of the brightening period after 1990. All models show dimming in the flattening period of the newly homogenized data. All models apart from CanESM5 show an averaged negative trend between the 6-year-means of 2003-2008 and 2009-2014, where the newly homogenized data show a brightening. A similar "jump" to the one seen in China can be identified slightly later in Japan (fig 2e). To our knowledge, we have no information of neither a replacement of instruments nor update in the classification of SDSR stations in Japan. Norris and Wild (2009) investigated the role of clouds for historical SDSR observations in China and Japan, and found the post-1990 brightening in Japan to be statistically significant, while the Chinese brightening was found nonsignificant. In this paper (published before Wang and Wild (2016)) half of the post-1990 brightening in China, and one third for Japan, was attributed to a reduction in cloud cover. These results point to a need for more studies assessing and evaluating available observational SDSR data. However, models do not accurately represent the strength of dimming throughout the whole period, nor the change in trend after 1990 and thus the time evolution of SDSR observed in China, with or without the early 1990s "jump" in brightening.

## 4.2 Aerosol effect on dimming

Out of all the experiments presented in Table 1 and Figure 3, only those containing anthropogenic aerosol emissions showed dimming. This is expected as aerosols have been presented as the main cause of reduction in SDSR in China by previous studies (Wild, 2009; Yunfeng et al., 2001; Kaiser and Qian, 2002).

Storelvmo et al. (2018) argues that the discrepancy seen between observed and modelled CMIP5 model mean global SDSR can be attained to errors in the treatment of processes that translate aerosol emissions into clear-sky and all-sky radiative forc-

ings. Here, we can see an anti-correlation between simulated $SO_4$ burdens from Figure 4(a) and (b), and simulated SDSR from Figure 2(b) and (f), respectively. Therefore we suggest that the simulated SDSR is dominantly a result of simulated $SO_4$ burdens. Simulated SDSR agrees relatively well with observed SDSR in Europe (Fig 2b), along with simulated $SO_4$ burden anti-correlating relatively well with observed SDSR in Europe (Fig 4a). This means that the model code translating burdens into SDSR in Europe can simulate changes in SDSR as a consequence of changed in aerosol emissions. If models translate burden into SDSR correctly in Europe, this does not necessarily mean that they translate burden into SDSR correctly in other regions. However, we suggest that the code translating burdens into SDSR should also work correctly in China, since also in China we find, that aerosols are the main cause of dimming, in agreement with (Wild, 2009; Yunfeng et al., 2001; Kaiser and Qian, 2002). Note also that we find no consistency between observed cloud cover changes, GEBA data and simulated cloud cover and SDSR anomalies in China (Table 2). By suggesting the translation process from burden to SDSR is behaving correct in both regions, the potential source of error causing discrepancies between observed and simulated SDSR can be traced to the causes of the simulated atmospheric burdens in the first place.

The sulfur dioxide emission inventory used as input for historical model simulations in CMIP6 is shown in Hoesly et al. (2018)(Figure 3), and the emissions as translated in four of the models of this study is shown in Figure A2.

Hoesly et al. (2018) have pointed to the need for emission uncertainties, but this has not been done for these emissions. Aas et al. (2019) have studied global and regional trends in atmospheric sulfur and found that uncertainties in emissions were largest in Asia, even if their study only went back to 1990.

Previous studies estimating $SO_2$ emissions include Lu et al. (2010), that found sulfur dioxide emissions in China increased by 53 % between 2000 and 2006 using technology based methodology, and thereby found similar results to that of Hoesly et al. (2018). Lu et al. (2010) also compared AOD derived SDSR to GEBA based SDSR data as shown in streets et al and found the GEBA based SDSR data to not accurately represent SDSR development in East Asia, this further underlines the need for more studies evaluating SDSR observations. Other studies such as Koukouli et al. (2018) have used satellite observations to estimate a new emission inventory for SO2 between 2005 and 2015 in China. We note that the year 2005 in Figure A2 is directly after the sharp increase in $SO_2$ emissions, and the biggest difference between the estimation made by Koukouli et al. (2018) and the $SO_2$ emission inventories in CMIP6 are a decrease in emissions after the year of 2011. This decrease in $SO_2$ emissions would intuitively result in a brightening, which is identified over the same time period in the homogenized data by Yang et al. (2018)(Fig 10 therein).

The modeled emissions of $SO_2$ as shown in Figure A2 over China showed no trace of a significant change in trend after 1990 in our observed SDSR timeseries as discussed in the previous section. Assuming sulfate burden is responsible for the observed multiyear trends of SDSR, we argue that errors in emissions inventories in China could be part of the problem.

## 5 Conclusions

Earlier studies have shown that previous generations of Earth System Models have not been able to reproduce the transient development of surface downwelling shortwave radiation (SDSR) in the last decades since 1960 when observations became available (Storelvmo et al. (2018), Allen et al. (2013)). This discrepancy is hypothesized to be related to increasing and then partially decreasing trends in global aerosol emissions and subsequent aerosol radiative effects, but the exact cause is unknown.

In this paper, we compared observations to model simulated surface downwelling shortwave radiation and cloud cover in specific regions for the time period 1961 to 2014. We found that in the *historical* experiments, CMIP6 models reproduce the transient development of SDSR well in Europe, but poorly in Asia. The multiple historical and associated perturbation experiments performed under CMIP6 reveal that only those simulations containing anthropogenic aerosol emissions show dimming, and the dimming is underestimated by most models. China exhibits a sharp positive trend in observed SDSR in the 1990s that is not found in historical model simulations. This "jump" has been suggested to be an artifact, but historical simulations also do not accurately represent the homogenized observed SDSR as proposed by Yang et al. (2018). We suggest that the continuous decrease in simulated SDSR is related to the continuous increase in atmospheric sulfate burden in the *historical* simulations over China. Following this logic, the observed transient development of SDSR points to the evolution of the sulfate burden in the models being wrong in this region. The sulfate burden is a result of sulfur dioxide emissions, gas-to-particle conversion and wet deposition. Sulfur dioxide emissions over China show neither sign of the observed trend change from gap filled GEBA data nor of the brightening-followed-flattening from the homogenized data as proposed by Yang et al. (2018). Sulfur dioxide emissions used in the models over China have a strong increase in the early 2000s, which can be observed as a sharp dimming at the same time in Figure 2(f). We suggest that the cause of the discrepancy between model and observations in transient SDSR in China is partly in erroneous emission inventories.

As the observed climate change is the result of warming from greenhouse gases and simultaneous cooling from aerosol radiative effects, getting aerosol emissions correct is important in earth system models.

Since the SDSR measurements are not only sensitive to aerosol effects, they might not be the most accurate way to infer historic aerosol loads and forcing. Further studies could include other observations and proxies for aerosol effects in the historical era, such as long-term satellite retrieved aerosol optical depth, deposition of anthropogenic sulfur, organic carbon and nitrate in ice cores, as well as daily temperature range records.

**Table 1.** Model participation, as used in this study, in CMIP6 model intercomparison projects (MIP) and their experiments.

| Experiment | NorESM2 | CanESM5 | MIROC6 | CESM2 | CNRM-ESM2-1 | GISS-E2-1-G | IPSL-CM6A-LR | MRI-ESM2-0 |
|---|---|---|---|---|---|---|---|---|
| historical | x | x | x | x | x | x | x | x |
| hist-aer | x | x | x | | | x | x | x |
| hist-GHG | x | x | x | x | | x | x | x |
| hist-nat | x | x | | x | | x | x | x |
| piClim-histaer | x | x | x | | | x | | |
| piClim-histall | x | x | x | | | x | x | |
| hist-piAer | x | | x | | | | | x |
| hist-piNTCF | x | | x | | x | | | x |
| histSST | x | | x | | x | | | x |

**Table 2.** Changes in Chinese cloud cover [%], all sky SDSR AS[W/m$^2$], and clear sky SDSR CS[W/m$^2$] between two 3-year means for three time periods. All model results are means made from three ensemble members of the historical simulation, colocated and extracted at Chinese GEBA station locations. Changes in cloud cover are from CRU gridded data and represent means colocated to chinese GEBA stations.

| Data | [1961-1963] — [1987-1989] | | | [1990-1992]—[1997-1999] | | | [2000-2002]—[2012-2014] | | |
|---|---|---|---|---|---|---|---|---|---|
| | [%] | AS[W/m$^2$] | CS[W/m$^2$] | [%] | AS[W/m$^2$] | CS[W/m$^2$] | [%] | AS[W/m$^2$] | CS[W/m$^2$] |
| NorESM2 | −1.0 | −4.6 | −4.0 | 0.6 | −1.0 | −0.4 | 0.3 | −3.9 | −5.0 |
| CanESM5 | −0.4 | −3.5 | −4.6 | −0.1 | 0.8 | 0.6 | −1.7 | −2.4 | −5.7 |
| MIROC6 | 0.4 | −4.4 | −3.6 | 1.2 | −1.3 | −0.4 | 0.5 | −5.5 | −5.3 |
| CESM2 | −1.0 | −2.6 | −3.6 | −0.2 | 0.0 | −0.2 | 0.0 | −5.3 | −6.7 |
| CNRM-ESM2-1 | −0.4 | −3.3 | −5.2 | −0.6 | 1.1 | −1.0 | −0.9 | −3.5 | −6.5 |
| GISS-E2-1-G | 1.3 | −3.7 | −6.4 | −0.2 | −0.7 | −1.2 | 2.5 | −8.7 | −9.9 |
| IPSL-CM6A-LR | −1.2 | −3.3 | −5.0 | 0.5 | −0.6 | −0.1 | −1.6 | −0.4 | −1.9 |
| MRI-ESM2-0 | −0.1 | −7.1 | −6.9 | −0.3 | −0.8 | −0.9 | −1.1 | −4.9 | −8.8 |
| **MODELMEAN** | −0.3 | −4.1 | −4.9 | 0.1 | −0.3 | −0.4 | −0.2 | −4.3 | −6.2 |
| **GEBA** | | −15.4 | | | 6.6 | | | 0.9 | |
| **CRU** | 0.1 | | | 3.0 | | | −1.0 | | |

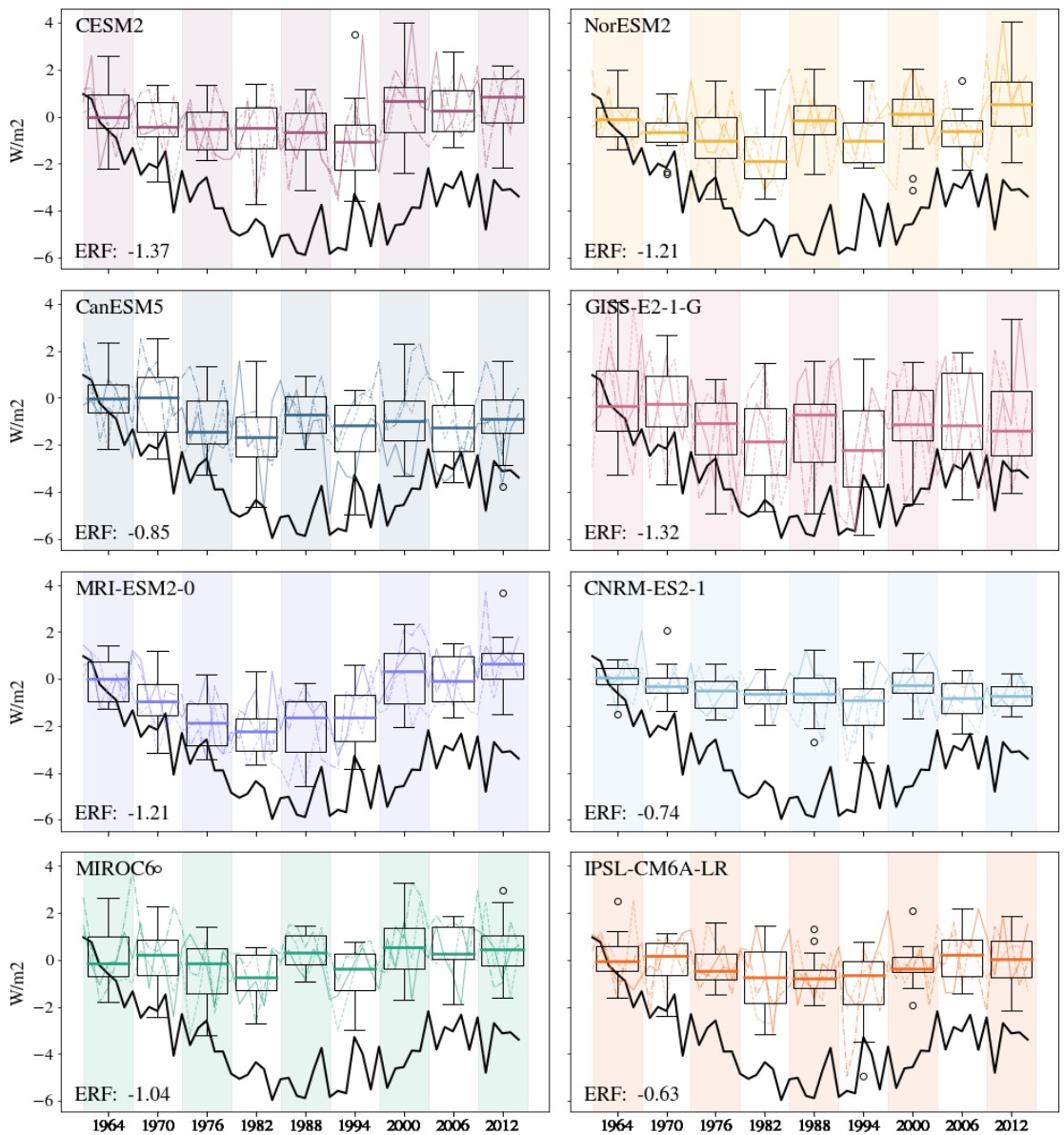

**Figure 1.** Global surface downwelling shortwave radiation (SDSR) anomaly at the surface for GEBA (black) and three ensemble members for the historical simulation of the eight models in this study. The boxes are made for 6-year intervals (shaded in background) based on 6 yearly means and three ensemble members per model. Colored lines behind boxes show yearly values of SDSR anomaly per ensemble member. The height of each box represents the interquartile range of the data, and the thick colored line within each box is the median. The whiskers show the minimum and maximum values of the selection of data, and the outliers are shown as a hollow dot. Results are co-located to all GEBA stations (1487) throughout the time period. The Aerosol ERF as found in Smith et al. (2020) per model is shown in the bottom left of each panel.

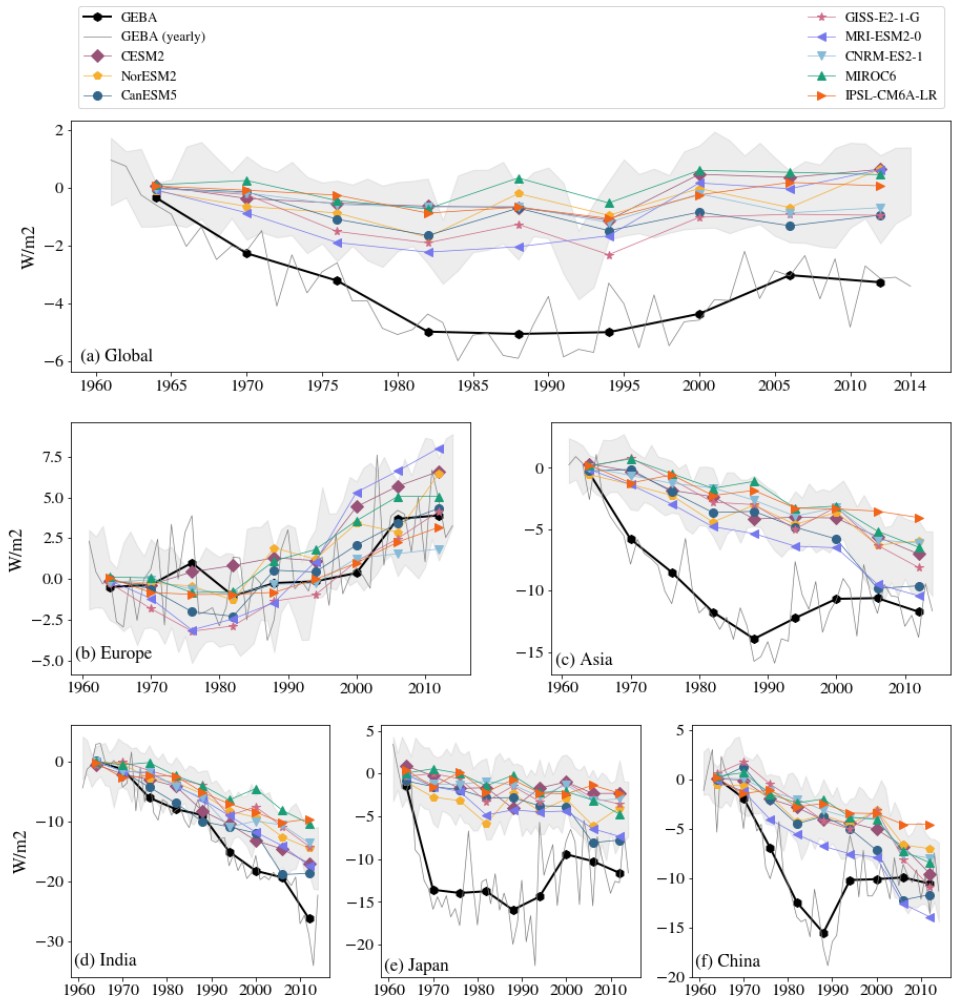

**Figure 2.** Six-year-averages of surface downwelling shortwave radiation (SDSR) anomaly at the surface for GEBA and eight earth system models. Results are co-located at (a) all GEBA stations (1487), (b) European (503), (c) Asian (311), (d), Indian (15), (e) Japanese (100), and (f) Chinese (119) stations. Numbers in parenthesis are number of ground stations in respective region. The entire 54-year period has beed divided into intervals of 6 years and then averaged together to make nine data points as shown by the markers. The grey shading represent one standard deviation from the yearly total ensemble mean.

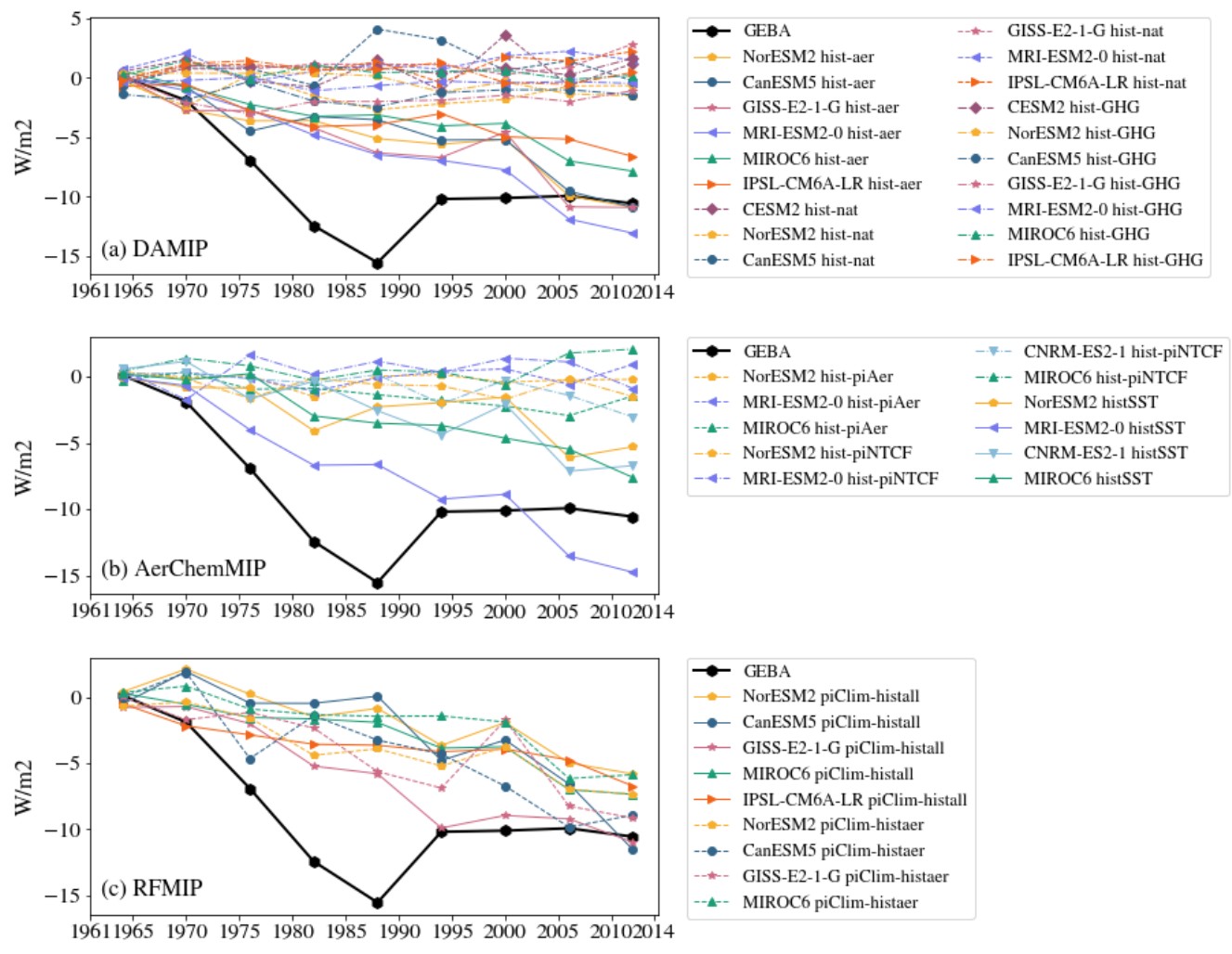

**Figure 3.** SDSR anomaly in China for all the CMIP6 simulations as listed in Table 1. All model results are co-located at GEBA station locations registered to China (119 stations). The entire 54-year period has beed divided into intervals of 6 years and then averaged together to make nine data points as shown by the markers.

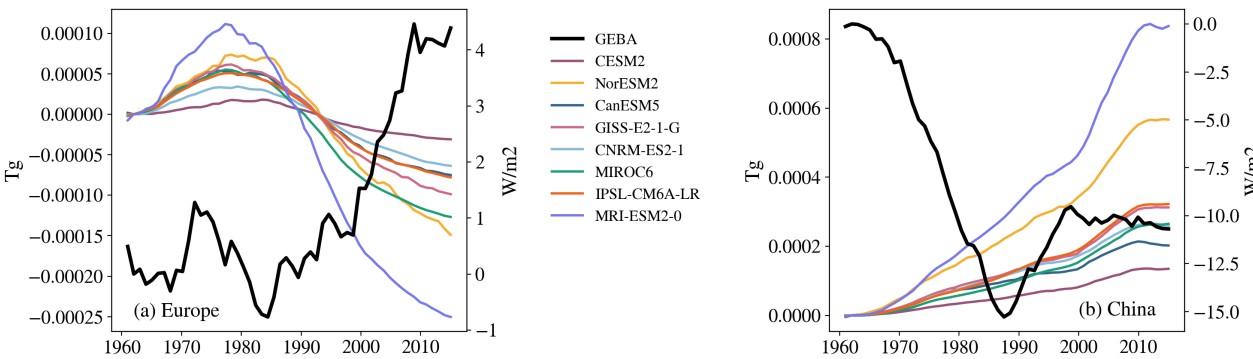

**Figure 4.** Anomaly of simulated atmospheric load of sulfate per model together with observed all sky SDSR anomaly in (a) Europe and (b) China. The GEBA data are shown as yearly anomalies, while the atmospheric loads have been smoothed using a 10-year running mean technique as explained in Section 2.3.

## Appendix A: Additional information

### A1 The European SDSR evolution

445 Figure A1 suggests cloud cover variation as a possible explanation of the local maximum in observed European SDSR during the period 1967-1978. Cloud cover exhibited a substantial minimum simultaneous to the maximum in SDSR. The peak is not reproduced in the historical runs of earth system models studied herein (see Figure 2(b)). Cloud cover variations that are not externally forced, but are rather a result of internal variability, cannot be expected to be reproduced in fully coupled earth system models. This might serve as an explanation why the substantial peak in SDSR between 1967 and 1978 is not reproduced

450 in the earth system models.

### A2 The data downloaded from ESGF

Table A2 shows an overview of the eight models used in this study. For the historical simulations three ensemble members per model was downloaded, with the variant labels r[1,2,3]i1p1f[1,2] for the variables rsds, rsdscs and clt. In addition the variable loadso4 and areacella was downloaded for one ensemble member per model in the historical simulation per model. In the

455 remaining experiments listed in Table 1 only one ensemble member per model was downloaded for the variable rsds, this was done as not every model provides more than one simulation per experiment.

### A3 Effects of cloud cover change on all sky SDSR

If we assume that $E_{\text{clear sky}}$ is the diurnal average clear sky SDSR in a region and that $\tau_{\text{cloud}}$ is the average cloud optical depth,

460 we can compute idealized effects of cloud changes on SDSR using the Beer-Lambert law:

$$E_{\text{surf}} = E_{\text{toa}} \exp(-\tau/\cos\phi),$$

where $\tau$ and $\phi$ denote optical depth and solar zenith angle, respectively. The change in SSR per $1\%$ change in cloud cover can then be computed:

$$\Delta E_{\text{surf per 1\%}} = 0.01 \times E_{\text{cloudy}} - E_{\text{clear sky}}$$

465

$$= 0.01 \times E_{\text{toa}} \exp(-\tau_{\text{cloud}}/\cos\phi + \ln(E_{\text{clear sky}}/E_{\text{toa}})) + 0.99 \times E_{\text{clear sky}} - E_{\text{clear sky}}$$

*Idealized computation for China:* Assuming that $\phi$ is between $30°$ and $70°$, that $E_{\text{clear sky}}$ is between 100 W/m$^2$ and 350 W/m$^2$ and that $E_{\text{toa}} =$1362 W/m$^2$ in China, the theoretical effect of $1\%$ increase in cloud cover on all sky SDSR is between -1 and -3.5 W/m$^2$, using the idealized computation described above.

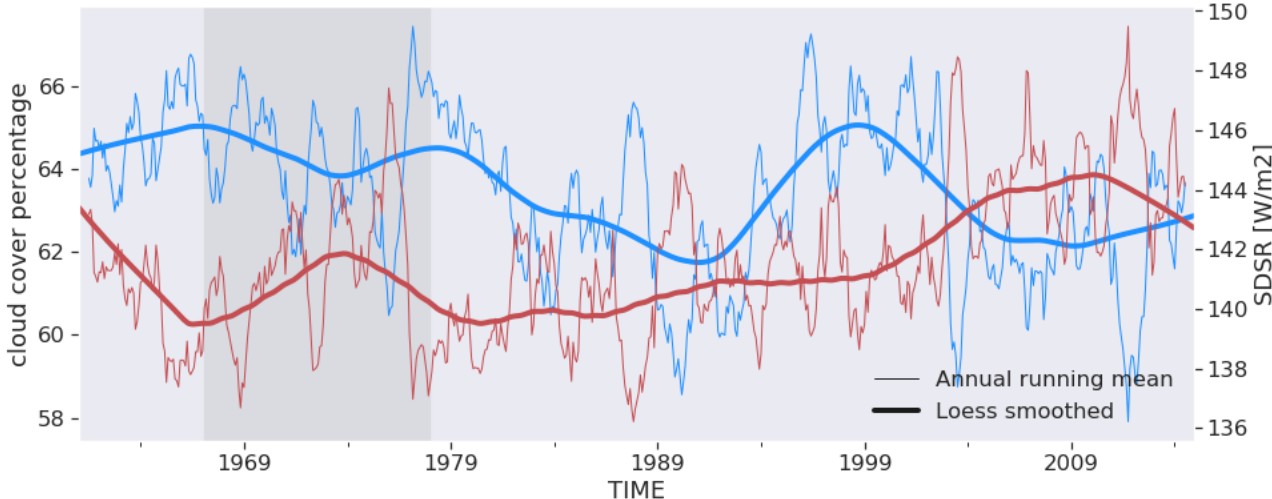

**Figure A1.** Timeseries of cloud cover (blue) and SDSR (red) between 1961 and 2014, co-located at GEBA sites in Europe. Thin lines show annual running means; bold lines show LOESS-smoothed variants. The shaded area delineates a period of interrupted dimming in Europe, between 1967 and 1978, which occurred simultaneous to a local minimum in the cloud cover trend.

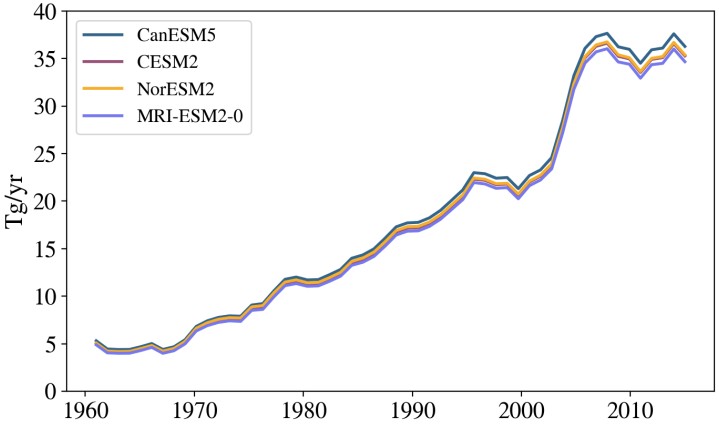

**Figure A2.** Emission of $SO_2$ in China, diagnosed by four of the models in this study. China is defined here as the area within latitudes [20°N–45°N], and longitudes [95°E–125°E].

**Table A1.** Global all sky SDSR and cloud cover averaged over the years 1961-1966 (baseline values) as observed (GEBA for radiation, CRU for cloud cover) and as simulated in the ensemble mean of three ensemble members in the historical experiment by each of the models of this study. Data from both CRU and models are retrieved after co-location to all GEBA sites.

| Model | SDSR [$W/m^2$] | Cloud Cover [%] |
|---|---|---|
| CESM2 | 186.3 | 63.9 |
| NorESM2 | 186.8 | 55.6 |
| CanESM5 | 189.5 | 56.2 |
| GISS-E2-1-G | 176.6 | 58.6 |
| MRI-ESM2-0 | 193.8 | 56.2 |
| CNRM-ES2-1 | 192.3 | 57.2 |
| MIROC6 | 184.3 | 50.4 |
| IPSL-CM6A-LR | 185.7 | 54.5 |
| **CRU** | | **57.4** |
| **GEBA** | **171.6** | |

**Table A2.** Details on the models used. IA: interactive aerosols NIA: not interactive aerosols.

| Institution | Model | Resolution | Aerosol module | Complexity | Reference |
|---|---|---|---|---|---|
| NCAR | CESM2 | 1.25x0.9 | MAM4 | IA | Danabasoglu et al. (2020) |
| CCCma | CanESM5 | 2.81x2.81 | CanAM4 | IA | Swart et al. (2019) |
| CNRM-CERFACS | CNRM-ESM2-1 | 1.4x1.4 | TACTIC_v2 | IA | Séférian et al. (2019) |
| IPSL | IPSL-CM6A-LR | 2.5x1.27 | INCA fields | NIA | Boucher et al. (2020) |
| NCC | NorESM2-LM | 2.5x1.875 | OsloAero6 | IA | Seland et al. (2020) |
| MRI | MRI-ESM2-0 | 1.125x1.125 | MASINGAR mk-2r4c | IA | Yukimoto et al. (2019) |
| MIROC | MIROC6 | 1.4x1.4 | SPRINTARS | IA | Tatebe et al. (2019) |
| NASA-GISS | GISS-E2-1-G | 2.5x2.0 | OMA fields | NIA | Kelley et al. (2020) |

*Author contributions.* KM wrote most of the article and did all analysis of CMIP6 data. MS and TS contributed to design of the study and helped editing the text. DO, PN, JC and TT, NO, SB, GG contributed model data via the ESGF CMIP6 archive. IJ and MW contributed with observational data, and IJ wrote part of the appendix. All co-authors contributed to the analysis and gave feedback to the manuscript.

*Competing interests.* No competing interests

*Acknowledgements.* This study benefited greatly from the CMIP6 data infrastructure for handling and providing model data for analysis. KM, MS and TS acknowledge funding from the European Union's Horizon 2020 project FORCeS under grant agreement No 821205. We acknowledge support from the Research Council of Norway funded project KeyClim (295046). Jan Griesfeller is thanked for data organisation. High performance computing and storage resources were provided by the Norwegian infrastructure for computational science (through projects NS2345K, NS9560K and NS9252K) and the Norwegian Meteorological Institute. TT was supported by the supercomputer system of the National Institute for Environmental Studies, Japan, and JSPS KAKENHI Grant Number JP19H05669. NO was supported by the Japan Society for the Promotion of Science (grant numbers: JP18H03363, JP18H05292, and JP20K04070), the Environment Research and Technology Development Fund (JPMEERF20172003, JPMEERF20202003, and JPMEERF20205001) of the Environmental Restoration and Conservation Agency of Japan, the Arctic Challenge for Sustainability II (ArCS II), Program Grant Number JPMXD1420318865, and a grant for the Global Environmental Research Coordination System from the Ministry of the Environment, Japan. GEBA is supported by the Federal Office of Meteorology and Climatology MeteoSwiss in the framework of GCOS Switzelrand. MW acknowledges fundings obtained from the Swiss National Science foundation grand No.$200020_1$88601.

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
