# Peer review of "Bias in CMIP6 models as compared to observed regional dimming and brightening"

_Atmospheric Chemistry and Physics, 2019_

## Referee Comment (RC1) · Anonymous Referee #1 · 28 Mar 2020

Moseid et al. compare surface downwelling shortwave radiation from CMIP6 models and from ground stations. They show the discrepancy between modeled and observed SDSR is partly caused by erroneous aerosol and aerosol precursor emission inventories, thus providing important information for the evaluation of ESM. While the research topic is essential, the methodology can be improved to clarify the impacts of clouds and cloud-aerosol interaction. Instead of using all-sky SDSR, I would suggest the authors compare the sunny-day SDSR from CMIP6 and from ground stations throughout the whole text. To be more accurate, I would also suggest the authors compare the SDSR conditions on the atmospheric relative humidity, which is associated with the scattering from water vapor. Note that the clear-sky SDSR from climate models is usually used

for calculating cloud radiative forcing and is not the same as sunny-day SDSR. Other comments: The title: I would not use the "1961-2014" in the title. It provides little information. The title: compare to -> compare with The title: maybe the authors should include "aerosol", which is the theme of the paper Figure 3: Please double check the cloud fraction and the calculation of anomaly. If the trend is reversed, it explains everything.

---

## Referee Comment (RC2) · Anonymous Referee #2 · 30 Mar 2020

**Manuscript Summary**

The manuscripts presents a comparison of surface downwelling shortwave radiation (SDSR) simulated by CMIP6 models against observations from the GEBA network over the period 1961-2014. The study attempts to provide additional observation constraint on CMIP6 models covering a period prior to the availability of other long-term ground based and satellite records. Evaluating CMIP6 models across Europe and China highlights important differences in the ability of models to simulate changes in the observed SDSR. The study attributes the failure of CMIP6 models to reproduce the temporal changes in SDSR over China to errors in the emissions of aerosols and their precursors and the translation into atmospheric burden. The paper represents an updated analysis using the CMIP6 models of that previously done with CMIP5 models. I think the paper presents analysis on an important topic which would be useful to the community, but I would like to see further details on the following points prior to publication.

**General comments**

1. It would improve the paper if more background information in the introduction section was provided on the key drivers of SDSR i.e. clouds and greenhouse gases can also influence SDSR in addition to aerosol effects.

2. Throughout the paper there are numerous mentions to the fact that aerosols play a key role in the dimming signal of SDSR observed and simulated across all regions. However, the same cannot be said for the observed brightening signal in more recent years. A key question seems to be why are aerosols a key driver in the dimming but not brightening? If the emission inventories and aerosols were in error throughout the whole period of study then surely the models would not be able to simulate the temporal evolution of both phenomenon across all regions?

3. The paper states that the CMIP6 models are able to represent the observed SDSR signal over Europe relatively well. However, I think there are a few interesting discrepancies which should be discussed further. Prior to 1980 the observations do not show much of a dimming signal (in fact the observed anomaly is slightly positive at times) but the CMIP6 models do show a consistent dimming signal. Is there a specific reason for the absence of a dimming in the observations, when we know there were large concentrations of aerosols over Europe at this time? Contrary to what was mentioned in point 2 above Europe is the only region where there is a simulated brightening signal in both the model and observations, implying that models are able to reproduce brightening signal over certain regions. It would be good to know if there a reason for this over Europe and does it occur over other regions like for example North America.

4. For the analysis over China the paper suggests that the error between the models and observations of SDSR are due to the errors in emission inventories that translate into errors in the calculation of atmospheric burden of aerosols. Are we certain that the errors in the emission inventories are that large to account for the discrepancy in model and observed SDSR? Is there an estimate of the uncertainty for the CMIP6 emission inventory and how does CMIP6 compare to other global and regional emission inventories? Can these differences explain some of the inconsistencies of models with observations? I am not convinced that the observed trend reversal in SDSR over China in ~1990 can be explained by errors in the emission inventories alone. Are we anticipating a slowing down of $SO_2$ emissions in China from the

1990s onwards? As far as I understand, anthropogenic emissions of aerosols and their precursors (particularly $SO_2$) have largely been increasing over China up until ~2010 when air pollutant control measures were then implemented to reduce emissions. Therefore, if aerosols were driving the temporal change in SDSR over China a dimming signal should have been observed up until this point, but it isn't. This is present in the observed and simulated change in SDSR over India but not China. How do this discrepancy match with the conclusions of the paper and what else could be driving the SDSR trend over China throughout this period? I think this needs to be explored further in the paper as the assumed underlying trend in emissions (and therefore aerosols) and SDSR do not seem to match over China and from what I can tell it cannot be reconciled by errors in the emission inventories alone.

5.  Only limited discussion within the paper is provided on clouds and aerosol-cloud interactions, which needs to be improved throughout the paper. Within section 3.3 a link is made between cloud cover change and SDSR but how much of an influence do clouds have on all-sky SDSR? How reliable are the observed and simulated cloud cover changes and can some uncertainty bounds be placed on them? Is a regional cloud cover change of 1-2% significant in terms of SDSR? In figure 3 the temporal change in observed cloud cover is similar to that in observed SDSR so even if clouds can't explain the magnitude and all of the observed change then surely they must be exerting some influence on SDSR? Is it possible to compare a clear-sky derived observed SDSR to that from model simulations to eliminate any influence of clouds on the signal?

6.  The previous comparison of CMIP5 models to observed SDSR by Storelvom et al., (2018) is mentioned throughout this study, with similar results presented here for CMIP6 models. A key question is therefore why has there been no improvement in simulating observed SDSR between CMIP5 and CMIP6 models? This is despite some changes to individual aerosol schemes within models and also different historical aerosol precursor emission datasets being used. Some discussion is needed on what is continually missing from the models and what are the model developments to focus on to improve the future simulation of SDSR.

7.  Further details are required, either in Table 1 or a new table, on each of the CMIP6 models used in this study. In particular, it would be useful to know horizontal resolution and some information on the individual chemistry and aerosol schemes in each model. This could provide useful information to the reader of the potential causes of discrepancies between models. In addition, it would be good to have a record somewhere of the actual output used from the ESGF (e.g. temporal period, variant ID, CMIP table ID etc). Furthermore, if there is data now available for additional CMIP6 models then it would be useful to include it, as long as it further informs the current study.

8.  The methods section (2.3) appears to lack important details of what model data is being used (see point 7) and how the gridded model data has actually been compared to the observations which are at point locations. In calculating the regional means at observation locations, do the number of sites used change over time period and does this have any impact on the results? Furthermore, in the results section the clear-sky SDSR is discussed but is not mentioned in the methods section. I also think that it is important to use multiple ensemble for meaning purposes when using coupled experiments members from models so that the internal variability in each model can be shown (this would give a range of variability important to show on some of the Figures for certain variables).

9. A General comment on the figures is that they could be improved to make them easier to read by using better colours (I found the light green very bright), tick marks on the axis and line types that are easier to distinguish between different model experiments. Also, if it is possible to include a measure of observational and model uncertainty on any of the figures then this would improve the comparisons. When values from figures are continually referred to in the text it would help the reader if there is reference table containing some of the key numbers included (like the supplementary table).

**Minor Comments**

Page 1, line 9 – Reword this sentence as mentioning SO2 emissions, which are not aerosols, and then other aerosols relevant to SDSR. Be more precise in this statement.

Page 1, line 13 – Can you say how much error is associated with aerosols and emission inventories that might contribute to error in SDSR?

Page 2, Line 30 – Is this statement true across all regions? What about for Europe?

Page 2, line 35 – For the introduction it would be good to include a bit more detail on what the GEBA observations on their own show before introducing any comparisons to models.

Page 2, line 46 – here the study says that two observational datasets are used but only one has been mentioned in the previous paragraph. Please include details of what is the second dataset used in this study.

Page 2, line 47 – please reword sentence "An explanation of the methods used to obtain and analyse the data complete Section 2."

Page 3, line 57 – it would be good to include the error in the observations on all figures to show the uncertainty in the observations.

Page 3, line 60 – Please clarify if this temporal gap filling technique allows for all 1487 stations to have a complete record of observations over the entire 1961-2014 and how this technique impacts the observations. If the number of stations used changes over the entire time period then it could be important for the analysis.

Page 3, line 74 – insert 'is' between "these the"

Page 4, line 93 – replace 'stales' with "stalls"

Page 4, line 94-95 – "So these experiments will show to what extent the removal of cloud cover change from global warming has an effect on SDSR." – I am sure that this is the case as there will be still be variability in the cloud fields simulated by climate models in these experiments. In addition, as the aerosol fields are changing in these experiments, they will also impact the simulated clouds in the models. Therefore, to make this statement further evidence would be required from each model that the cloud fields are being properly constrained to isolate their impacts on SDSR.

Page 4, line 107 – It would be good to show on a figure the spatial distribution of the GEBA observations within each defined region.

Page 4, line 110-112 – Please clarify exactly how anomalies have been calculated. Are anomalies calculated for each individual observation site within a region first before then calculating a regional mean value?

Page 4, line 112 - Supplementary table number is not shown

Page 4, line 113 – Provide more information on exactly what model data has been obtained from the ESGF (perhaps in a separate table) e.g. CMIP table ID, variant label etc. (see general comment 8)

Page 4, line 115 – I think it would be more prudent to use more ensemble members for coupled experiments and with this an idea of the internal variability for each model could be obtained for variables such as cloud cover and SDSR.

Page 4, line 116 – It is not clear if the 10-year running mean is used for the model data, observation data or both?

Page 5, line 121 – it is hard to see from Figure 1 a) as to whether the global SDSR representation in the models is similar to the observations at all. There is clearly a difference in magnitude but there does not appear to be a strong dimming signal in many of the models. Is this just the scale on the figure or is there not much change in the model at all? Can the Figure be improved in any way to make this easier to see?

Page 5, line 122 – Change "these discrepancy originate" to "this discrepancy originates"

Page 5, line 125 – More discussion on European model observational differences (see general comments point 3)

Page 5, line 135 – I think that this is only true for certain models as others seem to have opposite temporal changes compared to observations e.g. NorESM2.

Page 5, line 138 – It is hard to say without tick marks on the figures as to whether the end points in models are similar to the observations. For example, is a -10 Wm$^{-2}$ anomaly in 2014 from GEBA considered to be similar to a ~-6 Wm$^{-2}$ from NorESM2?

Page 5, line 140 – please explain what "temporal forcing evolution" means in this context.

Page 6, line 156-157 – does this imply that the greenhouse gases impact on SDSR over China throughout this period?

Page 6, line 157-158 – I am not sure this is true for all models. The temporal evolution of SDSR from CanESM5 seems quite different in the historical and piClim-histall but perhaps not so much in MIROC6.

Page 6, line 167 – Aerosols have a key role in dimming but not it appears brightening – why not? (see general comment 2)

Page 6, lines 168-169 – similar to point above in that there are differences between these simulations which don't appear to be the temporal driver of SDSR but perhaps can influence it? It would be good to show the actual differences between models in these simulations and what influence other factors (like clouds and greenhouse gases) can have on SDSR.

Page 6, line 173 – how has all-sky SDSR been decomposed into clear-sky?

Page 6, line 179-180 – Can the clear-sky and all-sky changes be shown on the same figure to compare differences?

Page 6, line 182-189 –How have the changes in model cloud cover been calculated? This needs to be in the methods section. Also line 183-184 states that cloud cover changes mask the clear-sky SDSR signal. This implies that the clear-sky decrease would have been even larger without changes to clouds indicating that clouds do have an important influence on SDSR in models. I think this needs to be explained more - see general comment section 5 for more details.

Page 7, line 193 – "session" should be "section"

Page 7, line 194 – "In this session we found the clear-sky SDSR to be stronger than all-sky SDSR, indicating the simulated dimming is primarily caused by aerosol-radiation interactions." But also that clouds have an influence on SDSR too.

Page 7, line 205 – "$SO_2$ burden" is mentioned but should this not be $SO_4$ burden.

Page 7, line 205-206 – Given that all models have the same $SO_2$ emissions, do we know why the changes in SO4 burden are so different between NorESM2 and CESM2? Could this indicate some of the potential problems in translating emissions into atmospheric burden or aerosols, which lead to errors in SDSR?

Page 7, line 210 – can a more scientific term be used than "real story".

Page 7, line 210-211 – This sentence makes the assumption that aerosols are the sole driving force in SDSR and that it is only the emissions and removal processes that could be in error. Other potential causes could be mentioned like the model translation of emissions to burden which leads to the larger differences in simulated $SO_4$ burdens between models. Also see major comments above.

Page 7, line 212 – "the precursor of $SO_2$", should this not be $SO_4$?

Page 7, line 215-218 – Should we be expecting a trend reversal in $SO_2$ emissions over China between 1980 and 1990? At this point in time emissions would have been increasing over China and emissions have only begun to reduce recently (since 2010). See general comment point 4

Page 8, line 235 – Is it possible to include the clear-sky proxy from GEBA here and compare to that from models on Figure 3 to show how well models simulate the aerosol radiation interactions?

Page 8, line 238 – change "shown in Figure displayed" to "(Fig. 2) show"

Page 8, line 242 – But the magnitude of the dimming was not sufficient to reproduce that observed (same as Allen?) and implies emissions are not high enough historically?

Page 8, line 247 - change "burden of $SO_2$" to "burden of $SO_4$".

Page 8, Lines 246-249 - The study only shows change in SDSR is opposite to $SO_4$ burden over Europe and not the case over China so can we really say that the process of translating burden to forcings are ok? What about over other regions? Might not just be due to errors in atmospheric burdens, but other factors combining?

Page 8, Line 250 – "The models of this study …" changed to "The models used in this study …"

Page 9, line 254-255 – Should we expect a reversal of emissions across China over this period?

Page 9, line 256 – Is this referring to Figure 3 in Hoesly et al., (2018)? Make clearer.

Page 9, line 258 – should we expect BC and OC to influences SDSR much? Need to mention these aerosol components earlier in the manuscript if going to mention now as no introduction to them at all. All discussion previously has been made about $SO_4$ so why suddenly bring them in now?

Page 9, line 259-261 – Do these studies give an uncertainty in emission inventories and can this be used to see if it can account for the differences between model and observed SDSR.

Page 9, line 270 – change "CMIP6 experiment models" to "experiments, CMIP6 models"

Page 9, line 273 – mention that the dimming is underestimated by the models.

Page 9, line 276-279 – Would we not have anticipated the $SO_4$ burden to have increased across China over this period as $SO_2$ emissions are anticipated to have also increased? Are the errors in $SO_4$ burden and $SO_2$ emissions really that large to account for the observed discrepancy in SDSR? More work to back up this statement and other factors should be included in conclusions. Uncertainty in emission inventories probably do contribute to this but the trend changes in SDSR and anticipated emission changes don't match for China, so this cannot be the sole reason and needs to be expanded on. see general comment point 4.

Page 10, line 285-287 – how would these future investigations improve our understanding of SDSR temporal evolution?

Fig. 2b – why is CanESMS so different in Hist-Nat and does show that other drivers influence the SDSR trend?

Fig. 3b – Can the uncertainty in cloud cover from observations and models be shown?

Fig. 4 – CESM2 seems to show a small change, can you confirm that this model has interact aerosols included? If not then why such a small change compared to others?

---

## Author Comment (AC1) · 31 Aug 2020

**Response to the reviewers**

We thank the reviewers for their critical assessment of our work. In the following we address their concerns point by point.
* * *
**Reviewer 1**

**General comments**

**Reviewer Point P 1.1** — Moseid et al. compare surface downwelling shortwave radiation from CMIP6 models and from ground stations. They show the discrepancy between modeled and observed SDSR is partly caused by erroneous aerosol and aerosol precursor emission inventories, thus providing important information for the evaluation of ESM. While the research topic is essential, the methodology can be improved to clarify the impacts of clouds and cloud-aerosol interaction. Instead of using all-sky SDSR, I would suggest the authors compare the sunny-day SDSR from CMIP6 and from ground stations throughout the whole text.

**Reply**: We agree that the manuscript should include a description of the impact of clouds and cloud-aerosol interactions. A new part was added in line 27-36 in the revised manuscript:
*Aerosol particles cause changes in the amount of sunlight reaching the surface together with changes in insolation , cloud cover, water vapor and other radiatively active gases (Wild et al., 2018). Insolation at the top of the atmosphere changes on millennial timescales when the Earth's orbital parameters change, and the famous 11-year cycle does not create multidecadal trends in surface radiation. Water vapor amount have not changed in a sufficient magnitude in recent decades to have an effect on decadal fluctuations in incoming sunlight at the surface (Wild (2009), Wang and Yang (2014), Yang et al. (2019), Hoyt and Schatten (1993), Ramanathan and Vogelmann (1997), Solomon et al. (2010)), and radiatively active gases dominate in the longwave spectrum (Ramanathan et al. (1989)). The relative roles of clouds, aerosols and their interactions in the variations of historically are still disputed, but previous studies have found that aerosol effects dominate on multidecadal timescales, while cloud effects are relevant on shorter timescales (Wild (2016),Romanou et al. (2007)).*

And Section 3.4 "Clear sky and cloud cover in China" has been improved throughout, including changing Figure 3 into a table that is easier to read and analyze.
In The discussion section we have added the lines 326-330:
*Analyzing the RFMIP simulations in light of Folini and Wild (2015) points to aerosol effects dominating over cloud cover for causing multi-year SDSR changes in China. As mentioned in the introduction, the relative roles of clouds, aerosols and their interactions in dimming and brightening are still disputed. Table 2 showed inconclusive connections between modelled and observed cloud cover, but clear connections between clear sky SDSR and all sky SDSR.*
Unfortunately neither GEBA nor CMIP6 models provide sunny day SDSR. Previous studies such as Allen et al. (2013) have used the GEBA data set to create a clear sky proxy for a selection of stations to compare with the clear sky flux variable of CMIP models. However, this is beyond the scope of our study.

**Reviewer Point P 1.2** — To be more accurate, I would also suggest the authors compare the SDSR conditions on the atmospheric relative humidity, which is associated with the scattering from water vapor.

Note that the clear-sky SDSR from climate models is usually used for calculating cloud radiative forcing and is not the same as sunny-day SDSR.

**Reply**: We are looking at longtime fluctuations in SDSR. Water vapor has not changed in a sufficient magnitude in recent decades to have an effect on decadal fluctuations in SDSR (Wang and Yang (2014), Yang et al. (2019), Wild (2009), Hoyt and Schatten (1993), Ramanathan and Vogelmann (1997), Solomon et al. (2010)). This was added in the new version of the manuscript in line 26-33 as cited above. We therefore assume in this study that the SDSR effects of water vapor scattering is negligible.

**Minor comments**

**Reviewer Point P 1.3** — The title: I would not use the "1961-2014" in the title. It provides little information.

**Reply**: Fixed.

**Reviewer Point P 1.4** — The title: compare to -> compare with.

**Reply**: Fixed.

**Reviewer Point P 1.5** — The title: maybe the authors should include "aerosol", which is the theme of the paper

**Reply**: Fixed.

**Reviewer Point P 1.6** — Figure 3: Please double check the cloud fraction and the calculation of anomaly. If the trend is reversed, it explains everything.

**Reply**: This is double checked and the presented Figure was correct. In the new version of the manuscript this Figure has been made into a table. (Table 2 in the revised version)
* * *
**Reviewer 2**

**General comments**

**Reviewer Point P 2.1** — It would improve the paper if more background information in the introduction section was provided on the key drivers of SDSR i.e. clouds and greenhouse gases can also influence SDSR in addition to aerosol effects.

**Reply**: We thank the reviewer for the comment and agree more background information should be provided regarding SDSR. We have added a more detailed description of what can influence SDSR in line 27-36 in the introduction, as cited in our reply to reviewer point P1.1.

**Reviewer Point P 2.2** — Throughout the paper there are numerous mentions to the fact that aerosols play a key role in the dimming signal of SDSR observed and simulated across all regions. However, the same cannot be said for the observed brightening signal in more recent years. A key question seems to be why are aerosols a key driver in the dimming but not brightening?
If the emission inventories and aerosols were in error throughout the whole period of study then surely the models would not be able to simulate the temporal evolution of both phenomenon across all regions?

**Reply**: We respond to this point in two parts - first the role of aerosols in brightening:
We would like to point the reviewer to the studies by Allen et al. (2013), Chiacchio et al. (2015) and Wild (2012), which show that indeed aerosols are a key driver to the observed brightening in recent years. The reduction of anthropogenic aerosol emission leads to brightening. We would also like to thank the reviewer for mentioning this point that we did not explicitly make in the original manuscript. The following text has now been added in line 45-48:
*In some areas a positive trend in SDSR follows the dimming, and this SDSR increase has been termed "brightening" (Wild et al., 2005). Brightening is connected to the reduction in anthropogenic aerosol emission (Nabat et al., 2014). Less particles suspended in the air allows for more sunlight to reach the surface and thereby an increase in the measured SDSR.*

Then the final question on emission inventories: Correct, this is why we are proposing errors in emission inventories as a possible reason for discrepancies in the regions where the models are not able to simulate the temporal evolution of dimming (and brightening).

**Reviewer Point P 2.3** — The paper states that the CMIP6 models are able to represent the observed SDSR signal over Europe relatively well. However, I think there are a few interesting discrepancies which should be discussed further. Prior to 1980 the observations do not show much of a dimming signal (in fact the observed anomaly is slightly positive at times) but the CMIP6 models do show a consistent dimming signal. Is there a specific reason for the absence of a dimming in the observations, when we know there were large concentrations of aerosols over Europe at this time? Contrary to what was mentioned in point 2 above Europe is the only region where there is a simulated brightening signal in both the model and observations, implying that models are able to reproduce brightening signal over certain regions. It would be good to know if there a reason for this over Europe and does it occur over other regions like for example North America.

**Reply**: The referee is right that Figure 1b does show some interesting discrepancies in the beginning of the time period in the study that was not mentioned in the original manuscript. The observational data set used in this study starts in 1961, and the anomaly shown in the figure is made as a difference from the the mean value of SDSR from 1961-1966. Since the European dimming started before 1960 (Wild, 2012) the "true" European SDSR anomaly might not be achieved using this data set, as is also seen by the weak European dimming in Storelvmo et al. (2018) using the same data set.
In the new version of the manuscript we have added in line 197-2000:
*The dimming in Europe is believed to have started before the start time (1961) of the observational*

*data set used here (Wild, 2009), which partly explains why the dimming in Figure 2 (b) is weak. GEBA shows a short-term positive anomaly between 1970 and 1980, which is not caught by the models. This peak is currently unexplained, but a short assessment of its possible association to changes in cloud cover is found in Section A2 the Appendix.*

The observed and simulated brightening in Europe are quite comparable and we therefore propose that the emission inventories of aerosols in Europe are estimated well.

North America has not been shown in Figure 1, but is included here in the reply as supplementary Figure S1. See Leibensperger et al. (2012b) and Leibensperger et al. (2012a) for a closer look at the climatic effects in North America due to anthropogenic aerosol emissions. We chose Europe and Asia as areas of focus to give the readers a clean impression of one example region where the models perform well and one example region where they do not perform well.

**Reviewer Point P 2.4** — For the analysis over China the paper suggests that the error between the models and observations of SDSR are due to the errors in emission inventories that translate into errors in the calculation of atmospheric burden of aerosols. (1)Are we certain that the errors in the emission inventories are that large to account for the discrepancy in model and observed SDSR? Is there an estimate of the uncertainty for the CMIP6 emission inventory and how does CMIP6 compare to other global and regional emission inventories? (2) Can these differences explain some of the inconsistencies of models with observations? I am not convinced that the observed trend reversal in SDSR over China in 1990 can be explained by errors in the emission inventories alone. (3) Are we anticipating a slowing down of SO2 emissions in China from the 1990s onwards? As far as I understand, anthropogenic emissions of aerosols and their precursors (particularly SO2) have largely been increasing over China up until 2010 when air pollutant control measures were then implemented to reduce emissions. Therefore, if aerosols were driving the temporal change in SDSR over China a dimming signal should have been observed up until this point, but it isn't. This is present in the observed and simulated change in SDSR over India but not China. (4) How do this discrepancy match with the conclusions of the paper and what else could be driving the SDSR trend over China throughout this period? I think this needs to be explored further in the paper as the assumed underlying trend in emissions (and therefore aerosols) and SDSR do not seem to match over China and from what I can tell it cannot be reconciled by errors in the emission inventories alone.

**Reply**: To make it easier for the reader we have marked up numbers to the questions in the reviewer point. (1) We are not certain that the errors in the emission inventories are large enough to account for the discrepancy in model and observed SDSR alone, but we suggest that this error plays an important role. Unfortunately there is no estimate of uncertainty in the CMIP6 emission inventories, but this is planned to be included in the next generation of CMIP emission inventories (see Hoesly et al. (2018)). Due to the lacking estimation of uncertainty we do not have evidence to say that errors in emission inventories are too small to cause a discrepancy between model and observation.

(2) The CMIP6 emission inventory is made using CEDS that makes datasets based on EDGAR, as is described in more detail in Hoesly et al. (2018). There are probably some differences between the CMIP6 data set and other regional emission data sets, but this study does not look further into finding such differences. We propose at least some of the discrepancy between model and observed SDSR is caused by errors in emission inventories, but we do not have enough evidence to claim that all discrepancies are emission driven. We recognize that the original manuscript may have given the wrong impression to the reader that errors in emission inventories *alone* cause all discrepancies, and this has now been

addressed and made clearer throughout the text.

(3) During the review process we found more information regarding the observed trend reversal in the GEBA data in China. According to the CMIP6 data set of sulfate emission we do not expect a slow down of emitted $SO_2$ from 1990, but rather from around 2005. The observed trend reversal in SDSR does therefore not fit with CMIP6 emitted sulfate. However, previous studies have found that the trend reversal in SDSR is to a considerable extent caused by the fact that the measurement devices at the Chinese radiation network stations were replaced with new ones between 1991 and 1993, which caused a spurious upward jump in the records (Wang and Wild (2016), Wang and Yang (2014), Yang et al. (2019)). With this new information we have added a Section 4.1 "The trend reversal in China" under Section 4 "Discussion" where we compare our results to that of Yang et al. (2018) where the "jump" has been removed by homogenization. The main point from this section can be summarized as in line 359-360:

*Models do not accurately represent the strength of dimming, or the evolutionary pattern of SDSR observed in China with or without the early 1990s brightening* (the "jump").

(4) The conclusions of the paper propose that errors in anthropogenic aerosol emission inventories play a role in the discrepancy seen between simulated and observed SDSR. Even if the trend reversal in the observed SDSR in China was to be an artifact, the models would still largely underestimate the magnitude of dimming. With regards to the trend reversal, the assumed underlying trend of increasing sulfate emission until 2010 as proposed by the reviewer (and CMIP6) is being questioned in this manuscript, as even though Wang and Wild (2016) suggests most of the "jump" is an artifact, they still estimate that 20% is real. We thank the reviewer for this comment and hope the new Section 4.1 is satisfactory.

**Reviewer Point P 2.5** — Only limited discussion within the paper is provided on clouds and aerosol-cloud interactions, which needs to be improved throughout the paper. Within section 3.3 a link is made between cloud cover change and SDSR but how much of an influence do clouds have on all-sky SDSR? How reliable are the observed and simulated cloud cover changes and can some uncertainty bounds be placed on them? Is a regional cloud cover change of 1-2% significant in terms of SDSR? In figure 3 the temporal change in observed cloud cover is similar to that in observed SDSR so even if clouds can't explain the magnitude and all of the observed change then surely they must be exerting some influence on SDSR? Is it possible to compare a clear-sky derived observed SDSR to that from model simulations to eliminate any influence of clouds on the signal?

**Reply**: We thank the reviewer for this comment, and would refer to our reply to reviewer 1 point 1 (P1.1) where we cite lines from our new manuscript regarding clouds' role in all sky SDSR. In the appendix of the new version of the manuscript (Section A2) we have added an idealized estimation for how much a theoretical 1 % cloud cover increase in China would affect SDSR. Line 435-436:

*... in China, the theoretical effect of 1% increase in cloud cover on all sky SDSR is between -1 and -3.5 W/m$^2$, using the idealized computation described above.*

Previous studies have found that the link between cloud cover and SDSR trends depends on what region you are looking into. In Europe cloud cover has most of an effect on SDSR on the shorter time scales, and the dimming and following brightening observed in Europe is dominantly caused by changes in anthropogenic aerosol emission and thereby the aerosol absorption and scattering (Norris and Wild, 2007). In China cloud cover made a negligible contribution to all sky SDSR trend in GEBA until 1989. After 1980 the heavily discussed trend reversal is observed in China, and Norris and Wild (2009) suggests half of the observed brightening between 1990 and 2002 is caused by a reduction in cloud cover. Please note that this paper was published before the proposal of the trend reversal being an artifact of a change

in instrumentation (Wang and Wild, 2016). This complicates the story and is the reason Norris and Wild (2009) is not discussed in the new version of our manuscript.

In North America cloud cover is found to have played an important role in the observed brightening (Long et al., 2018). Other studies have made clear-sky derived observed SDSR (Norris and Wild (2007), Norris and Wild (2009)) when assessing the cloud signal for Europe and China (mentioned above in this reply), but this goes beyond the scope of our study.

**Reviewer Point P 2.6** — The previous comparison of CMIP5 models to observed SDSR by Storelvmo et al., (2018) is mentioned throughout this study, with similar results presented here for CMIP6 models. A key question is therefore why has there been no improvement in simulating observed SDSR between CMIP5 and CMIP6 models? This is despite some changes to individual aerosol schemes within models and also different historical aerosol precursor emission datasets being used. Some discussion is needed on what is continually missing from the models and what are the model developments to focus on to improve the future simulation of SDSR.

**Reply**: To answer this question we must first find out whether the source of the error is within the model's codes or within the emission inventories, or a combination. Storelvmo et al. (2018) argues that the discrepancy between observed and modelled SDSR may be attributed to errors in the treatment of processes that translate aerosol emissions into clear-sky and all-sky radiative forcings. Here, we show that simulated SDSR develops similarly in time, but opposite in sign, to simulated atmospheric burden of SO2. By doing this we narrow down the potential source of error by suggesting that the atmospheric burden in the models are at fault, and that the processes translating burden into clear-sky and all-sky radiative forcings are behaving as expected. The final answer of what is at fault is still not found, but we suggest to have found a piece of the puzzle in the emission inventories.

It is important to note that Storelvmo et al. (2018) included all CMIP5 models, and we "only" include eight models.

We thank the reviewer for this comment and have updated the end of the conclusion in the new manuscript. Lines 409-414 is added:

*As the observed climate change is the result of warming from greenhouse gases and simultaneous cooling from aerosol radiative effects, getting aerosol emissions correct is an important part in earth system models' ability to simulate the past for the right reasons.*

*Since the SDSR measurements are not only sensitive to aerosol effects, further studies could include other observations and proxies for aerosol effects in the historical era, such as long-term satellite retrieved aerosol optical depth, deposition of anthropogenic sulphur, organic carbon and nitrate in ice cores, as well as daily temperature range records.*

**Reviewer Point P 2.7** — Further details are required, either in Table 1 or a new table, on each of the CMIP6 models used in this study. In particular, it would be useful to know horizontal resolution and some information on the individual chemistry and aerosol schemes in each model. This could provide useful information to the reader of the potential causes of discrepancies between models. In addition, it would be good to have a record somewhere of the actual output used from the ESGF (e.g. temporal period, variant ID, CMIP table ID etc). Furthermore, if there is data now available for additional CMIP6 models then it would be useful to include it, as long as it further informs the current study.

**Reply**: We thank the reviewer for this comment and we have added Table A2 in the Appendix of the paper listing information such as variant ID, variables, references to model documentation, horizontal

resolution and aerosol scheme. More data has been published since the first submission of this paper, and we have therefore decided to include more models in this study. Three models have been added (GISS-E2-1-G, IPSL-CM6A-LR, MRI-ESM2-0) to the analysis as more data was released.

**Reviewer Point P 2.8** — The methods section (2.3) appears to lack important details of what model data is being used (see point 7) and how the gridded model data has actually been compared to the observations which are at point locations. In calculating the regional means at observation locations, do the number of sites used change over time period and does this have any impact on the results? Furthermore, in the results section the clear-sky SDSR is discussed but is not mentioned in the methods section. I also think that it is important to use multiple ensemble for meaning purposes when using coupled experiments members from models so that the internal variability in each model can be shown (this would give a range of variability important to show on some of the Figures for certain variables).

**Reply**: We thank the reviewer for this comment and agree that the methods section was indeed lacking both clarity and details. In Section 2.3 "Methods" in the new manuscript we have added line 138-140:
*All model output and CRU results have been co-located to GEBA station locations using the nearest neighbour method. This entails that if two GEBA stations are within one grid box of a model, data from that grid box will be retrieved twice by nearest neighbour interpolation, as every station has been weighted equally.*
We tackle the question regarding number of sites used in time in both Section 2.1 " Observations" with line 79-81 added in the new manuscript:
*This allows for all 1487 stations to have data on each time step, so that all regions have a complete record and the same amount of stations throughout the entire time period in question.*
And in Section 2.3 "Methods" in line 135-136:
*The number of stations per region remains constant throughout the time period.*
In the new version of the manuscript we have added three ensemble members per model for the historical simulation. Both inter-annual variability and inter-ensemble variability is shown in Figure 1 of the new manuscript, that is presented in the new section in results called Section 3.1 " Model variability". We have changed Figure 1 in the old manuscript into Figure 2 of the new manuscript, where we present ensemble means per model, and show shading for the standard deviations of the total 24 ensemble members.

**Reviewer Point P 2.9** — A General comment on the figures is that they could be improved to make them easier to read by using better colours (I found the light green very bright), tick marks on the axis and line types that are easier to distinguish between different model experiments. Also, if it is possible to include a measure of observational and model uncertainty on any of the figures then this would improve the comparisons. When values from figures are continually referred to in the text it would help the reader if there is reference table containing some of the key numbers included (like the supplementary table).

**Reply**: We thank the reviewer and have chosen a different color chart for the figures, more tick marks, and different line types to better differ the graphs. Variability and uncertainty is shown in the new Figure 1 and Figure 2 as explained in the previous reply (P2.8). We are currently not referring to specific values until Section 3.4 "Clear sky SDSR and cloud cover in China" where we have changed the previous Figure 3 into a table to make the point more clear and the discussion easier to follow.

**Minor comments**

**Reviewer Point P 2.10** — Page 1, line 9 – Reword this sentence as mentioning SO2 emissions, which are not aerosols, and then other aerosols relevant to SDSR. Be more precise in this statement.

**Reply**: Changed to line 11: *The emissions of $SO_2$ used in the models show no pattern that could explain the observed SDSR evolution over China, and neither do emission of aerosols relevant to SDSR, such as black carbon (BC).*

**Reviewer Point P 2.11** — Page 1, line 13 – Can you say how much error is associated with aerosols and emission inventories that might contribute to error in SDSR?

**Reply**: Unfortunately the emission inventory data set for CMIP6 does not have estimates of uncertainty, which is why we chose the word "partly" in line 13 as we have no evidence telling us how much of the discrepancy can be attributed to emission estimates.

**Reviewer Point P 2.12** — Page 2, Line 30 – Is this statement true across all regions? What about for Europe?

**Reply**: This statement is only true globally based on previous studies. Added the word global in line 48: *Previous studies show that historical simulations from ESMs do not reproduce the global transient development of SDSR as observed (Storelvmo et al. (2018), Wild (2009)).*

**Reviewer Point P 2.13** — Page 2, line 35 – For the introduction it would be good to include a bit more detail on what the GEBA observations on their own show before introducing any comparisons to models.

**Reply**: We thank the reviewer for pointing this out, as the introduction to global dimming mentioned several citations that all used GEBA to identify dimming (and regional brightening), which was not explicitly mentioned. This has now been clarified in the text in line 53-56: *this study we use gap-filled data based the GEBA dataset. The GEBA dataset is the observational dataset as used in the citations in the previous paragraph, together with several recent CMIP6 historical model experiments from eight ESMs to investigate the aerosol effect in the time period 1961-2014, globally and regionally.*

**Reviewer Point P 2.14** — Page 2, line 46 – here the study says that two observational datasets are used but only one has been mentioned in the previous paragraph. Please include details of what is the second dataset used in this study.

**Reply**: The second observational data set has been added in line 58-59: *We also use observational cloud cover data to briefly assess the role of cloud cover in the historical development of SDSR.*

**Reviewer Point P 2.15** — Page 2, line 47 – please reword sentence "An explanation of the methods used to obtain and analyse the data complete Section 2."

**Reply**: Changed to line 66-67: *The methods used to obtain and analyse the data finalize Section 2.*

**Reviewer Point P 2.16** — Page 3, line 57 – it would be good to include the error in the observations on all figures to show the uncertainty in the observations.

**Reply**: Unfortunately sources of error in observation differs from station to station and we only have a general estimation of error from the instruments used. In addition to the instrumental error presented in line 76 we have chosen to include a light grey line with the highly variable yearly observational data in the background of Figure 2 in the new version of the manuscript.

**Reviewer Point P 2.17** — Page 3, line 60 – Please clarify if this temporal gap filling technique allows for all 1487 stations to have a complete record of observations over the entire 1961-2014 and how this technique impacts the observations. If the number of stations used changes over the entire time period then it could be important for the analysis.

**Reply**: This has been clarified and is cited in reply to P2.8.

**Reviewer Point P 2.18** — Page 3, line 74 – insert 'is' between "these the"

**Reply**: Fixed.

**Reviewer Point P 2.19** — Page 4, line 93 – replace 'stales' with "stalls"

**Reply**: Fixed.

**Reviewer Point P 2.20** — Page 4, line 94-95 – "So these experiments will show to what extent the removal of cloud cover change from global warming has an effect on SDSR." – I am sure that this is the case as there will be still be variability in the cloud fields simulated by climate models in these experiments. In addition, as the aerosol fields are changing in these experiments, they will also impact the simulated clouds in the models. Therefore, to make this statement further evidence would be required from each model that the cloud fields are being properly constrained to isolate their impacts on SDSR.

**Reply**: We are not stating that all cloud cover change is removed, only the cloud cover change that is induced by global warming - as global warming essentially is removed in these experiments. Cloud cover will change due to aerosol emissions and thereby impact SDSR - but not due to global warming. Changed the wording to line 115: *So these piClim-experiments will show to what extent the removal of cloud cover change caused by global warming has an effect on SDSR.*

**Reviewer Point P 2.21** — Page 4, line 107 – It would be good to show on a figure the spatial distribution of the GEBA observations within each defined region.

**Reply**: Storelvmo et al. (2018)s Figure 1 is an excellent figure showing the spatial distribution of the stations used in both this and her study in addition to the trends of the stations in colours. I have added a reference to that figure in line 136.

**Reviewer Point P 2.22** — Page 4, line 110-112 – Please clarify exactly how anomalies have been calculated. Are anomalies calculated for each individual observation site within a region first before then calculating a regional mean value?

**Reply**: Clarified. New line 142-146: *When a result is shown as an anomaly, as opposed to an absolute value, the general formula has been to subtract the mean of the first five years of the investigated time*

*period (1961-2014) from the timeseries in question. To clarify - first an average value per year per region is calculated, and then a new mean is created from the first five years of this timeseries. This 5-year-mean is then subtracted from each year in the timeseries for the region in question and presented as an anomaly.*

**Reviewer Point P 2.23** — Page 4, line 112 - Supplementary table number is not shown

**Reply**: Fixed.

**Reviewer Point P 2.24** — Page 4, line 113 – Provide more information on exactly what model data has been obtained from the ESGF (perhaps in a separate table) e.g. CMIP table ID, variant label etc. (see general comment 8)

**Reply**: We thank the reviewer for this request and a table has been added as cited in the reply to P2.7

**Reviewer Point P 2.25** — Page 4, line 115 – I think it would be more prudent to use more ensemble members for coupled experiments and with this an idea of the internal variability for each model could be obtained for variables such as cloud cover and SDSR.

**Reply**: Three ensemble member have been used in the historical experiment, see reply to P2.8 for citation.

**Reviewer Point P 2.26** — Page 4, line 116 – It is not clear if the 10-year running mean is used for the model data, observation data or both?

**Reply**: Running means have been exchanged for 6-year-intervals means in most figures in the new manuscript. the only exception if Figure 4 which shows $SO_4$ burdens form models as a 10-year running mean, while the observation is shown as yearly data. This is clarified in the figure caption.

**Reviewer Point P 2.27** — Page 5, line 121 – it is hard to see from Figure 1 a) as to whether the global SDSR representation in the models is similar to the observations at all. There is clearly a difference in magnitude but there does not appear to be a strong dimming signal in many of the models. Is this just the scale on the figure or is there not much change in the model at all? Can the Figure be improved in any way to make this easier to see?

**Reply**: Figure 1a) corresponds to Figure 2a) in the new manuscript. The models generally do not represent the *global* change in SDSR as observed. We have included gray shading for the ensemble standard deviations and changed the method from a running mean to 6-year-interval-means to show clearly the weak signal in the models in Figure 2 of the new manuscript.

**Reviewer Point P 2.28** — Page 5, line 122 – Change "these discrepancy originate" to "this discrepancy originates"

**Reply**: Fixed.

**Reviewer Point P 2.29** — Page 5, line 125 – More discussion on European model observational differences (see general comments point 3)

**Reply**: This discussion has been added and is cited in reply to P2.3.

**Reviewer Point P 2.30** — Page 5, line 135 – I think that this is only true for certain models as others seem to have opposite temporal changes compared to observations e.g. NorESM2.

**Reply**: We agree and this line has been removed.

**Reviewer Point P 2.31** — Page 5, line 138 – It is hard to say without tick marks on the figures as to whether the end points in models are similar to the observations. For example, is a -10 Wm-2 anomaly in 2014 from GEBA considered to be similar to a  -6 Wm-2 from NorESM2?

**Reply**: We agree that this statement was questionable in the old manuscript. By adding more models to the analysis the remark of similar end points between model and observations became blatantly wrong and we have removed all statements regarding this.

**Reviewer Point P 2.32** — Page 5, line 140 – please explain what "temporal forcing evolution" means in this context.

**Reply**: This line has been removed due to the added discussion of the trend reversal in China in observations.

**Reviewer Point P 2.33** — Page 6, line 156-157 – does this imply that the greenhouse gases impact on SDSR over China throughout this period?

**Reply**: When adding more models to the study this implication became untrue, and the statement has been removed in the new manuscript.

**Reviewer Point P 2.34** — Page 6, line 157-158 – I am not sure this is true for all models. The temporal evolution of SDSR from CanESM5 seems quite different in the historical and piClim-histall but perhaps not so much in MIROC6.

**Reply**: With new models added to the study the entire RFMIP paragraph has been updated to line 239-244: *Recall that the experiments of RFMIP utilize pre-industial SST's, meaning essentially there is no global warming in these experiments. In the RFMIP experiments shown in Figure 3(c) both piClim-histaer and piClim-histall contain anthropogenic aerosol emissions, and all simulations show a continuous dimming throughout the period. There is no clear divide between experiments containing both GHG emissions in addition to anthropogenic aerosol emissions (solid lines, piClim-histall) and the experiments only containing anthropogenic aerosol emissions (stipled lines, piClim-histaer). This implies that greenhouse gases without their global warming effect do not largely affect all sky SDSR throughout the period.*

**Reviewer Point P 2.35** — Page 6, line 167 – Aerosols have a key role in dimming but not it appears brightening – why not? (see general comment 2)

**Reply**: We thank the reviewer for this comment and refer til our reply in P2.2

**Reviewer Point P 2.36** — Page 6, lines 168-169 – similar to point above in that there are differences between these simulations which don't appear to be the temporal driver of SDSR but

perhaps can influence it? It would be good to show the actual differences between models in these simulations and what influence other factors (like clouds and greenhouse gases) can have on SDSR.

**Reply**: Clouds and greenhouse gases can influence SDSR, but are, as mentioned in the introduction, not a dominant driver of multidecadal SDSR changes. It is therefore expected to see small differences between these simulations. The overall picture of models showing dimming with anthropogenic aerosol emissions, and no dimming without it remains whether or not you include greenhouse gases or SST changes. This has been clarified in line 245-249: *Overall there is a clear difference in SDSR between experiments that include anthropogenic aerosol emissions and experiments that do not. Dimming is apparent in every simulation containing anthropogenic aerosol emissions, but absent in the simulations containing pre-industrial aerosols only. This points to anthropogenic aerosol emissions playing a key role in global dimming. Whether the sea surface temperature is pre-industrial, prescribed historical, or decided by a coupled ocean model seems to be unimportant for the SDSR in most models.*

**Reviewer Point P 2.37** — Page 6, line 173 – how has all-sky SDSR been decomposed into clear-sky?

**Reply**: This is a diagnostic that is output from the models. The general idea is that clear-sky SDSR from models represents the amount of sunlight reaching the surface if all shortwave effects from clouds were removed. Clear-sky SDSR is not to be confused with sunny day SDSR which is from actual cloud free days.

**Reviewer Point P 2.38** — Page 6, line 179-180 – Can the clear-sky and all-sky changes be shown on the same figure to compare differences?

**Reply**: This figure has been replaced by Table 2 in the new manuscript showing changes in cloud cover, all sky SDSR and clear sky SDSR for three different time periods.

**Reviewer Point P 2.39** — Page 6, line 182-189 –How have the changes in model cloud cover been calculated? This needs to be in the methods section. Also line 183-184 states that cloud cover changes mask the clear-sky SDSR signal. This implies that the clear-sky decrease would have been even larger without changes to clouds indicating that clouds do have an important influence on SDSR in models. I think this needs to be explained more - see general comment section 5 for more details.

**Reply**: Cloud cover is a standard output from climate models and has not been calculated by the authors, and the source of the data has been added in the Appendix of the new manuscript. The effect of clouds in SDSR has been added and is cited in our reply to P2.5.

**Reviewer Point P 2.40** — Page 7, line 193 – "session" should be "section"

**Reply**: Fixed.

**Reviewer Point P 2.41** — Page 7, line 194 – "In this session we found the clear-sky SDSR to be stronger than all-sky SDSR, indicating the simulated dimming is primarily caused by aerosol-radiation interactions." But also that clouds have an influence on SDSR too.

**Reply**: This sentence has been removed in the renewal of the manuscript. But in general clouds have an influence on SDSR which is clarified and cited in our reply to P2.5. but our findings are that aerosols effects are the dominating cause of dimming.

**Reviewer Point P 2.42** — Page 7, line 205 – "SO2 burden" is mentioned but should this not be SO4 burden.

**Reply**: Fixed.

**Reviewer Point P 2.43** — Page 7, line 205-206 – Given that all models have the same SO2 emissions, do we know why the changes in SO4 burden are so different between NorESM2 and CESM2? Could this indicate some of the potential problems in translating emissions into atmospheric burden or aerosols, which lead to errors in SDSR?

**Reply**: Burdens are a result of emission, aerosol formation, transport and deposition. The emissions in both models are the same but the other processes dependent on many different parameterisations within each model. The atmospheric circulation in CESM2 and NoreSM2 differs, among other things, so for example a sulfate particle may be brought higher up in the atmosphere in NorESM2 - giving sulfate a longer lifetime and thereby making NorESM2 have a higher sulfate burden. In addition to this these burdens are calculated using co-location to point locations as described in the methods section, and this is where transportation plays a role.

**Reviewer Point P 2.44** — Page 7, line 210 – can a more scientific term be used than "real story".

**Reply**: Definitely. Added line 307: *Assuming GEBA data provide a reasonable representation of the historical development of SDSR and implicitly sulphur burdens in China, the problem in $SO_4$ burden must come from either the emissions, aerosol formation, transport or the removal processes of $SO_4$.*

**Reviewer Point P 2.45** — Page 7, line 210-211 – This sentence makes the assumption that aerosols are the sole driving force in SDSR and that it is only the emissions and removal processes that could be in error. Other potential causes could be mentioned like the model translation of emissions to burden which leads to the larger differences in simulated SO4 burdens between models. Also see major comments above.

**Reply**: As cited in the previous answer we have added "aerosol formation, transport" in this sentence. The model burden is translated from emission, transportation and removal processes. We are assuming the model translation from emission to burden is behaving as expected in the two regions of special interest, and an explanation for this is found in the Dicussion in line 366-377:
*..we can see an anti-correlation between simulated $SO_4$ burdens from Figure 4 (a) and (b), and simulated SDSR from Figure 2 (b) and (f), respectively. Therefore we suggest that the simulated SDSR is dominantly a result of simulated $SO_4$ burdens. Since simulated SDSR agree relatively well with observed SDSR in Europe (Fig 2(b)), simulated $SO_4$ burden anti-correlates relatively well with observed SDSR in Europe as well (Fig 4 (a)). This means that the model code translating burdens into forcing in Europe is behaving as expected. Cloud cover affects forcing, and if models translate burden into forcing correctly in Europe, does not mean they translate burden correctly in other regions. We suggest the code translating burdens into forcing in China is also behaving as expected, due to the above statement*

*saying aerosols are the main cause of dimming in China (Wild, 2009; Yunfeng et al., 2001; Kaiser and Qian, 2002), and the lack of consistency found in simulated cloud cover and SDSR anomalies in China (Table 2). By suggesting the translation process from burden to forcing is behaving as expected in both regions, the potential source of error causing discrepancies between observed and simulated SDSR can be traced to the causes of the simulated atmospheric burdens in the first place.* If there is an error in burden than the error is sourced in either emissions, transportation or removal processes.

**Reviewer Point P 2.46** — Page 7, line 212 – "the precursor of SO2", should this not be SO4?

**Reply**: Fixed.

**Reviewer Point P 2.47** — Page 7, line 215-218 – Should we be expecting a trend reversal in SO2 emissions over China between 1980 and 1990? At this point in time emissions would have been increasing over China and emissions have only begun to reduce recently (since 2010). See general comment point 4

**Reply**: A section discussing the trend reversal was added and is cited in our reply to P2.4.

**Reviewer Point P 2.48** — Page 8, line 235 – Is it possible to include the clear-sky proxy from GEBA here and compare to that from models on Figure 3 to show how well models simulate the aerosol radiation interactions?

**Reply**: Unfortunately that is not easily done and is beyond the scope of this study. There is currently an NSF project working on creating clear sky proxies at ETH Zurich lead by Dr Martin Wild.

**Reviewer Point P 2.49** — Page 8, line 238 – change "shown in Figure displayed" to "(Fig. 2) show"

**Reply**: Fixed.

**Reviewer Point P 2.50** — Page 8, line 242 – But the magnitude of the dimming was not sufficient to reproduce that observed (same as Allen?) and implies emissions are not high enough historically?

**Reply**: Correct.

**Reviewer Point P 2.51** — Page 8, line 247 - change "burden of SO2" to "burden of SO4".

**Reply**: Fixed.

**Reviewer Point P 2.52** — Page 8, Lines 246-249 - The study only shows change in SDSR is opposite to SO4 burden over Europe and not the case over China so can we really say that the process of translating burden to forcings are ok? What about over other regions? Might not just be due to errors in atmospheric burdens, but other factors combining?

**Reply**: A new section answering this question has been added in the Dicussion and has been cited in our reply to P2.45.

**Reviewer Point P 2.53** — Page 8, Line 250 – "The models of this study ..." changed to "The models used in this study ..."

**Reply**: Fixed.

**Reviewer Point P 2.54** — Page 9, line 254-255 – Should we expect a reversal of emissions across China over this period?

**Reply**: A section discussing the trend reversal was added and is cited in our reply to P2.4.

**Reviewer Point P 2.55** — Page 9, line 256 – Is this referring to Figure 3 in Hoesly et al., (2018)? Make clearer.

**Reply**: Fixed.

**Reviewer Point P 2.56** — Page 9, line 258 – should we expect BC and OC to influences SDSR much? Need to mention these aerosol components earlier in the manuscript if going to mention now as no introduction to them at all. All discussion previously has been made about SO4 so why suddenly bring them in now?

**Reply**: This sentence has been removed from the manuscript. With future studies a line have been added in line 393-394: *In addition model experiments looking into the effects of black carbon versus sulfate on all sky SDSR would greatly benefit our understanding of global and regional dimming and brightening.*

**Reviewer Point P 2.57** — Page 9, line 259-261 – Do these studies give an uncertainty in emission inventories and can this be used to see if it can account for the differences between model and observed SDSR.

**Reply**: Unfortunately none of these studies presents number for uncertainty in emission inventories, but Aas et al. (2019) show annual average trend in sulfate in aerosols from 2000-2015 and found that the standard deviation was larger than the actual trend for East Asia, and none of the locations used in that study was located in China Aas et al. (2019)[Tab. 1].

**Reviewer Point P 2.58** — Page 9, line 270 – change "CMIP6 experiment models" to "experiments, CMIP6 models"

**Reply**: Fixed.

**Reviewer Point P 2.59** — Page 9, line 273 – mention that the dimming is underestimated by the models.

**Reply**: Fixed.

**Reviewer Point P 2.60** — Page 9, line 276-279 – Would we not have anticipated the SO4 burden to have increased across China over this period as SO2 emissions are anticipated to have also increased? Are the errors in SO4 burden and SO2 emissions really that large to account for the observed discrepancy in SDSR? More work to back up this statement and other factors should

be included in conclusions. Uncertainty in emission inventories probably do contribute to this but the trend changes in SDSR and anticipated emission changes don't match for China, so this cannot be the sole reason and needs to be expanded on. see general comment point 4.

**Reply**: As we do not know the estimation uncertainty for emission we do not have evidence to rule out that the emission inventories can have large errors. The observed trend reversal in China have a new discussion section which is cited in our reply to P2.4.

**Reviewer Point P 2.61** — Page 10, line 285-287 – how would these future investigations improve our understanding of SDSR temporal evolution?

**Reply**: A comparison between different observational datasets such as GEBA and ice cores will give a unique insight in aerosol presence before the satellite era, especially if the emission inventories are wrong. Satellite products can be used to compare to CRU cloud cover data and give a full picture. Model experiments with different aerosol emission as input can disentangle the role of different aerosols on SDSR assuming their translation from emission to SDSR is correct. Have added in the manuscript "Since the SDSR measurements are not only sensitive to aerosol effects, further studies could include... "

**Reviewer Point P 2.62** — Fig. 2b – why is CanESMS so different in Hist-Nat and does show that other drivers influence the SDSR trend?

**Reply**: We do not know why CanESM5 differs from the others in it's hist-nat experiment, but this single experiment is unfortunately not enough evidence to say that other drivers influence the SDSR trend - except for in CanESM5 only.

**Reviewer Point P 2.63** — Fig. 3b – Can the uncertainty in cloud cover from observations and models be shown?

**Reply**: The section explaining the CRU dataset has been improved in line 78-84:
*CRU covers the period 1901-2017 (Harris et al., 2020) and consists of a climatology made from measurements at meteorological stations around the globe, interpolated to a 0.5° latitude/longitude resolution grid covering continental areas. Information on interpolation methods and procedures used to create the gridded data set are given in Harris et al. (2020) and references therein. In short, CRU has its foundation in station data, but is interpolated to a grid using angular-distance weighting. The cloud cover variable is largely derived as a secondary variable, based on measurements of other parameters such as sunshine hours and diurnal temperature range.*
As cloud cover is a secondary observed variable we have added line 274 in the Results section regarding clear sky and cloud cover data in China:
*It is important to note that the robustness of observed cloud cover changes must be verified by satellite observations, which goes beyond the scope of this study.*
Uncertainty in cloud cover from models is hard to quantify with only three ensemble members, but Table A1 shows the different baseline values for cloud cover in each model, which can be seen with a spread of 50%-64%.

**Reviewer Point P 2.64** — Fig. 4 – CESM2 seems to show a small change, can you confirm that this model has interact aerosols included? If not then why such a small change compared to others?

**Reply**: Aerosols interact with the climate in CESM2. We have added a Table in the appendix (Table A2) showing which models that have interactive aerosols and which that don't.

**References**

Aas, Wenche, Augustin Mortier, Van Bowersox, Ribu Cherian, Greg Faluvegi, Hilde Fagerli, Jenny Hand, Zbigniew Klimont, Corinne Galy-Lacaux, Christopher M. B. Lehmann, Cathrine Lund Myhre, Gunnar Myhre, Dirk Olivié, Keiichi Sato, Johannes Quaas, P. S. P. Rao, Michael Schulz, Drew Shindell, Ragnhild B. Skeie, Ariel Stein, Toshihiko Takemura, Svetlana Tsyro, Robert Vet, and Xiaobin Xu (2019), "Global and regional trends of atmospheric sulfur." *Scientific Reports*, 9, 953, URL `https://www.nature.com/articles/s41598-018-37304-0`.

Allen, R. J., J. R. Norris, and M. Wild (2013), "Evaluation of multidecadal variability in CMIP5 surface solar radiation and inferred underestimation of aerosol direct effects over Europe, China, Japan, and India." *Journal of Geophysical Research: Atmospheres*, 118, 6311–6336, URL `https://agupubs.onlinelibrary.wiley.com/doi/abs/10.1002/jgrd.50426`.

Chiacchio, Marc, Fabien Solmon, Filippo Giorgi, Paul Stackhouse, and Martin Wild (2015), "Evaluation of the radiation budget with a regional climate model over Europe and inspection of dimming and brightening." *Journal of Geophysical Research: Atmospheres*, 120, 1951–1971, URL `https://agupubs.onlinelibrary.wiley.com/doi/full/10.1002/2014JD022497`.

Folini, D. and M. Wild (2015), "The effect of aerosols and sea surface temperature on China's climate in the late twentieth century from ensembles of global climate simulations." *Journal of Geophysical Research: Atmospheres*, 120, 2261–2279, URL `https://agupubs.onlinelibrary.wiley.com/doi/abs/10.1002/2014JD022851`.

Harris, Ian, Timothy J. Osborn, Phil Jones, and David Lister (2020), "Version 4 of the CRU TS monthly high-resolution gridded multivariate climate dataset." *Scientific Data*, 7, 109, URL `https://www.nature.com/articles/s41597-020-0453-3`. Number: 1 Publisher: Nature Publishing Group.

Hoesly, Rachel M., Steven J. Smith, Leyang Feng, Zbigniew Klimont, Greet Janssens-Maenhout, Tyler Pitkanen, Jonathan J. Seibert, Linh Vu, Robert J. Andres, Ryan M. Bolt, Tami C. Bond, Laura Dawidowski, Nazar Kholod, June-ichi Kurokawa, Meng Li, Liang Liu, Zifeng Lu, Maria Cecilia P. Moura, Patrick R. O'Rourke, and Qiang Zhang (2018), "Historical (1750–2014) anthropogenic emissions of reactive gases and aerosols from the Community Emissions Data System (CEDS)." *Geoscientific Model Development*, 11, 369–408, URL `https://www.geosci-model-dev.net/11/369/2018/`.

Hoyt, Douglas V. and Kenneth H. Schatten (1993), "A discussion of plausible solar irradiance variations, 1700-1992." *Journal of Geophysical Research: Space Physics*, 98, 18895–18906, URL `https://agupubs.onlinelibrary.wiley.com/doi/abs/10.1029/93JA01944`. _eprint: https://agupubs.onlinelibrary.wiley.com/doi/pdf/10.1029/93JA01944.

Kaiser, Dale P. and Yun Qian (2002), "Decreasing trends in sunshine duration over China for 1954–1998: Indication of increased haze pollution?" *Geophysical Research Letters*, 29, 38–1–38–4, URL `https://agupubs.onlinelibrary.wiley.com/doi/abs/10.1029/2002GL016057`.

Leibensperger, E. M., L. J. Mickley, D. J. Jacob, W.-T. Chen, J. H. Seinfeld, A. Nenes, P. J. Adams, D. G. Streets, N. Kumar, and D. Rind (2012a), "Climatic effects of 1950–2050 changes in US anthropogenic aerosols – Part 1: Aerosol trends and radiative forcing." *Atmospheric Chemistry and Physics*, 12, 3333–3348, URL `https://www.atmos-chem-phys.net/12/3333/2012/`. Publisher: Copernicus GmbH.

Leibensperger, E. M., L. J. Mickley, D. J. Jacob, W.-T. Chen, J. H. Seinfeld, A. Nenes, P. J. Adams, D. G. Streets, N. Kumar, and D. Rind (2012b), "Climatic effects of 1950–2050 changes in US anthropogenic aerosols – Part 2: Climate response." *Atmospheric Chemistry and Physics*, 12, 3349–3362, URL `https://www.atmos-chem-phys.net/12/3349/2012/acp-12-3349-2012.html`. Publisher: Copernicus GmbH.

Long, C. N., E. G. Dutton, J. A. Augustine, W. Wiscombe, M. Wild, S. A. McFarlane, and C. J. Flynn (2018), "Significant decadal brightening of downwelling shortwave in the continental United States." *Journal of Geophysical Research: Atmospheres*, URL `https://agupubs.onlinelibrary.wiley.com/doi/full/10.1029/2008JD011263%4010.1002/%28ISSN%2921`
Publisher: John Wiley & Sons, Ltd.

Nabat, Pierre, Samuel Somot, Marc Mallet, Arturo Sanchez-Lorenzo, and Martin Wild (2014), "Contribution of anthropogenic sulfate aerosols to the changing Euro-Mediterranean climate since 1980." *Geophysical Research Letters*, 41, 5605–5611, URL `https://agupubs.onlinelibrary.wiley.com/doi/abs/10.1002/2014GL060798`.

Norris, Joel R. and Martin Wild (2007), "Trends in aerosol radiative effects over Europe inferred from observed cloud cover, solar "dimming," and solar "brightening"." *Journal of Geophysical Research: Atmospheres*, 112, URL `https://agupubs.onlinelibrary.wiley.com/doi/abs/10.1029/2006JD007794`. _eprint: https://agupubs.onlinelibrary.wiley.com/doi/pdf/10.1029/2006JD007794.

Norris, Joel R. and Martin Wild (2009), "Trends in aerosol radiative effects over China and Japan inferred from observed cloud cover, solar "dimming," and solar "brightening"." *Journal of Geophysical Research: Atmospheres*, 114, URL `https://agupubs.onlinelibrary.wiley.com/doi/abs/10.1029/2008JD011378`. _eprint: https://agupubs.onlinelibrary.wiley.com/doi/pdf/10.1029/2008JD011378.

Ramanathan, V., R. D. Cess, E. F. Harrison, P. Minnis, B. R. Barkstrom, E. Ahmad, and D. Hartmann (1989), "Cloud-radiative forcing and climate: results from the Earth radiation budget experiment." *Science*, 243, 57–63.

Ramanathan, V. and Andrew M. Vogelmann (1997), "Greenhouse Effect, Atmospheric Solar Absorption and the Earth's Radiation Budget: From the Arrhenius-Langley Era to the 1990s." *Ambio*, 26, 38–46, URL `https://www.jstor.org/stable/4314548`. Publisher: [Springer, Royal Swedish Academy of Sciences].

Romanou, A., B. Liepert, G. A. Schmidt, W. B. Rossow, R. A. Ruedy, and Y. Zhang (2007), "20th century changes in surface solar irradiance in simulations and observations." *Geophysical Research Letters*, 34, URL

https://agupubs.onlinelibrary.wiley.com/doi/abs/10.1029/2006GL028356. _eprint: https://agupubs.onlinelibrary.wiley.com/doi/pdf/10.1029/2006GL028356.

Solomon, Susan, Karen H. Rosenlof, Robert W. Portmann, John S. Daniel, Sean M. Davis, Todd J. Sanford, and Gian-Kasper Plattner (2010), "Contributions of Stratospheric Water Vapor to Decadal Changes in the Rate of Global Warming." *Science*, 327, 1219–1223, URL https://science.sciencemag.org/content/327/5970/1219. Publisher: American Association for the Advancement of Science Section: Research Article.

Storelvmo, Trude, Ulla K. Heede, Thomas Leirvik, Peter C. B. Phillips, Philipp Arndt, and Martin Wild (2018), "Lethargic Response to Aerosol Emissions in Current Climate Models." *Geophysical Research Letters*, 0, URL https://agupubs.onlinelibrary.wiley.com/doi/abs/10.1029/2018GL078298.

Wang, Y. W. and Y. H. Yang (2014), "China's dimming and brightening: evidence, causes and hydrological implications." *Ann. Geophys.*, 32, 41–55, URL https://www.ann-geophys.net/32/41/2014/.

Wang, Yawen and Martin Wild (2016), "A new look at solar dimming and brightening in China: CHINA DIMMING AND BRIGHTENING REVISITED." *Geophys. Res. Lett.*, 43, 11,777–11,785, URL http://doi.wiley.com/10.1002/2016GL071009.

Wild, Martin (2009), "Global dimming and brightening: A review." *Journal of Geophysical Research*, 114, URL http://doi.wiley.com/10.1029/2008JD011470.

Wild, Martin (2012), "Enlightening Global Dimming and Brightening." *Bull. Amer. Meteor. Soc.*, 93, 27–37, URL https://journals.ametsoc.org/doi/abs/10.1175/BAMS-D-11-00074.1.

Wild, Martin (2016), "Decadal changes in radiative fluxes at land and ocean surfaces and their relevance for global warming." *Wiley Interdisciplinary Reviews: Climate Change*, 7, 91–107, URL https://onlinelibrary.wiley.com/doi/abs/10.1002/wcc.372.

Wild, Martin, Hans Gilgen, Andreas Roesch, Atsumu Ohmura, Charles N. Long, Ellsworth G. Dutton, Bruce Forgan, Ain Kallis, Viivi Russak, and Anatoly Tsvetkov (2005), "From Dimming to Brightening: Decadal Changes in Solar Radiation at Earth's Surface." *Science*, 308, 847–850, URL https://science.sciencemag.org/content/308/5723/847. Publisher: American Association for the Advancement of Science Section: Report.

Wild, Martin, Maria Z. Hakuba, Doris Folini, Patricia Dörig-Ott, Christoph Schär, Seiji Kato, and Charles N. Long (2018), "The cloud-free global energy balance and inferred cloud radiative effects: an assessment based on direct observations and climate models." *Clim Dyn*, URL https://doi.org/10.1007/s00382-018-4413-y.

Yang, Su, Xiaolan L. Wang, and Martin Wild (2018), "Homogenization and Trend Analysis of the 1958–2016 In Situ Surface Solar Radiation Records in China." *J. Climate*, 31, 4529–4541, URL http://journals.ametsoc.org/doi/10.1175/JCLI-D-17-0891.1.

Yang, Su, Xiaolan L. Wang, and Martin Wild (2019), "Causes of Dimming and Brightening in China Inferred from Homogenized Daily Clear-Sky and All-Sky in

situ Surface Solar Radiation Records (1958–2016)." *J. Climate*, 32, 5901–5913, URL http://journals.ametsoc.org/doi/10.1175/JCLI-D-18-0666.1.

Yunfeng, Luo, Lu Daren, Zhou Xiuji, Li Weiliang, and He Qing (2001), "Characteristics of the spatial distribution and yearly variation of aerosol optical depth over China in last 30 years." *Journal of Geophysical Research: Atmospheres*, 106, 14501–14513, URL https://agupubs.onlinelibrary.wiley.com/doi/abs/10.1029/2001JD900030.

**1   Supplementary Material**

[Figure]

Figure S1: SDSR anomaly North America, model results are co-located to GEBA stations following the methodology as described in Moseid et al 2020 in prep

---

## Referee Report (RR1)

**Further Comments on "Bias in CMIP6 models as compared to observed regional dimming and brightening" manuscript**

The authors have thoroughly revised the manuscript and it now appears much improved than before, providing a more comprehensive assessment of observed SDSR trends and those simulated by CMIP6 models. I have provided a few detailed comments on the discussion of the new section 4.1 below and a few more minor comments further down on the revised manuscript.

I found the new section 4.1 very interesting, in that most (80%) of the observed positive trend in SDSR over China in the 1990s is down to instrument error. This "jump" in the 1990s also occurs in the Japan dataset (Fig. 2e) so I wonder is this could be due to the same issue here? The error in Chinese observations has been discussed in 4.1 but having read this and looking at Figure 10 in the Yang et al., (2018) paper I think more emphasis needs to be made in the text that the trend reversal is mostly an instrumental error. The trend reversal is consistently mentioned in other sections of the text and perhaps too much emphasis is still made that the models are not capturing this trend, even though we now know the observations presented over China in this period are mostly in error. There are also more similarities between the temporal trend in the $SO_2$ emissions and the new homogenised dataset e.g. increasing emissions up to ~2005, which coincides with the minimum SDSR in the Yang et al., (2018) homogenised data.

However, given the above I do not believe that this finding would change the main outcome of this paper much in that CMIP6 models underestimate the observed dimming signal over Asia and are unable to reproduce the recent brightening potentially due to uncertainties in the emission inventories. However, I think the direction of and timing of the temporal trends in observed and simulated SDSR over China (and Asia) would possibility match up better than with those observations currently presented in the manuscript i.e. the new observations show a dimming signal until ~2005 and then brightening. I was wondering if there is anyway to include the dataset from Yang et al., (2018) on any of the figures in the manuscript to provide a more direct comparison of the new and old observations with the model simulations? Also, I think some minor changes to text to include more reference to this observational change would be beneficial. This could go some way to addressing the current comparison for China (and also for Asia in Fig 2c), where the obvious discrepancy in the figures is the sharp change in observations in the 1990s, which we now know was not really observed.

As mentioned above the CMIP6 models would still underestimate the magnitude of the dimming signal across China and do not represent the observed brightening from ~2005, highlighting that there are uncertainties in the magnitude and timing of emissions over China. Therefore, potentially including some brief discussion on the recent literature of emission estimates of $SO_2$ over China, particular for recent years, would prove beneficial and aid the conclusions of the paper regarding $SO_2$, $SO_4$ and SDSR (e.g. Lu et al., 2010; Koukouli et al., 2018; Zheng et al., 2018).

Below are a few minor comments for the authors to consider.

A general comment (and mentioned specifically below) is to consider making more reference in the results to the fact that aerosols influence multi-decadal trends in SDSR and other variables (clouds) are more important for short term variability.

Within the revised manuscript there appear to be references to Figures, tables and Appendix items that need to be updated. For example, the results section refers to a Figure 5 but I have not seen a

Figure 5 in the revised manuscript. Also there does not appear to be a reference to Figure A1 in the text.

Page 2 line 31-33 – Not sure I understood this sentence properly, could do with clarifying a bit.

Page 2 lines 39-41 – Could link to the updated IPCC definitions, aerosol radiation interactions (Ari) and aerosol-cloud interactions (Aci).

Page 5 line 140 – I think hist-piNTCF does not fix $CH_4$ at pre-industrial levels so this might need to be removed from the description.

Page 7, line 213 – remove 'the' before 1961

Page 8 line 229 – remove 'between'

Page 9 line 258 – perhaps refer to multidecadal dimming signals

Page 9 line 261 – perhaps include 'changes in' before anthropogenic aerosol emissions'

Page 9, line 274-275 – perhaps again include reference to 'multidecadal trends in' all sky SDSR

Page 9, line 283-284 – but could this be due to the errors in the observations pointed out in Section 4.1?

Page 10 line 310-311 – Potentially remove/reword 'not by changing cloud cover' as could be misleading. Yes aerosols change the brightness of clouds but what about aerosol the effect on cloud lifetime?

Page 11 lines 229 – The calculation of the impact of changes in cloud cover on SDSR is presented but not really discussed. A simple sentence on the implications of this finding would improve its usefulness.

 Page 11 line 236-237 – Based on the all-sky and clear-sky results can you say then that the dimming is primarily caused by direct aerosol radiation interacts, with a smaller impact from aerosol-cloud interactions?

Page 11 line 340 – should you not specifically mention sulphate aerosols here, as not all aerosol scatter shortwave radiation?

Page 12 line 358-359 – Based on section 4.1 can we say that GEBA, as it is presented in Figure 4b provides a reasonable representation of the historical development of SDSR over China?

Page 15 line 458 – should we expect a trend reversal in emissions given that this was mostly identified as an observation error in section 4.1?

Appendix A3 Line 518 – makes reference to rsds twice.

**References**

Koukouli, M. E., Theys, N., Ding, J., Zyrichidou, I., Mijling, B., Balis, D., and van der A, R. J.: Updated $SO_2$ emission estimates over China using OMI/Aura observations, Atmos. Meas. Tech., 11, 1817–1832, https://doi.org/10.5194/amt-11-1817-2018, 2018

Lu, Z., Streets, D. G., Zhang, Q., Wang, S., Carmichael, G. R., Cheng, Y. F., Wei, C., Chin, M., Diehl, T., and Tan, Q.: Sulfur dioxide emissions in China and sulfur trends in East Asia since 2000, Atmos. Chem. Phys., 10, 6311–6331, https://doi.org/10.5194/acp-10-6311-2010, 2010.

Yang, S. U. and Wang, X. L.: Homogenization and Trend Analysis of the 1958-2016 In Situ Surface Solar Radiation Records in China, , doi:10.1175/JCLI-D-17-0891.1, n.d.

Zheng, B., Tong, D., Li, M., Liu, F., Hong, C., Geng, G., Li, H., Li, X., Peng, L., Qi, J., Yan, L., Zhang, Y., Zhao, H., Zheng, Y., He, K., and Zhang, Q.: Trends in China's anthropogenic emissions since 2010 as the consequence of clean air actions, Atmos. Chem. Phys., 18, 14095–14111, https://doi.org/10.5194/acp-18-14095-2018, 2018.

---

## Author Response (AR2)

**Author response**

We thank the reviewer for her/his thorough assessment of our work. The following changes have been made to the manuscript:
- The term trend reversal has been removed and changed to post-1990 trend change and emphasis on this trend change has been reduced.
- We have expanded Section 4.1 in the manuscript to express the need for more studies evaluating observational SDSR datasets.
- Added a brief discussion around Norris and Wild (2009) in regards to the japanese post-1990 brightening.
- Added some of the references suggested by the reviewer in Section 4.2 regarding sulfur dioxide emissions.

In the following we address the reviewers concerns point by point.
* * *
**Reviewer 1**

**General comments**

**Reviewer Point P 1.1** — The authors have thoroughly revised the manuscript and it now appears much improved than before, providing a more comprehensive assessment of observed SDSR trends and those simulated by CMIP6 models. I have provided a few detailed comments on the discussion of the new section 4.1 below and a few more minor comments further down on the revised manuscript.
I found the new section 4.1 very interesting, in that most (80%) of the observed positive trend in SDSR over China in the 1990s is down to instrument error. This "jump" in the 1990s also occurs in the Japan dataset (Fig. 2e) so I wonder is this could be due to the same issue here? The error in Chinese observations has been discussed in 4.1 but having read this and looking at Figure 10 in the Yang et al., (2018) paper I think more emphasis needs to be made in the text that the trend reversal is mostly an instrumental error. The trend reversal is consistently mentioned in other sections of the text and perhaps too much emphasis is still made that the models are not capturing this trend, even though we now know the observations presented over China in this period are mostly in error. There are also more similarities between the temporal trend in the SO2 emissions and the new homogenised dataset e.g. increasing emissions up to 2005, which coincides with the minimum SDSR in the Yang et al., (2018) homogenised data.

**Reply**: We thank the reviewer for their comment. We agree that "trend reversal" has been still too strong a statement and we have reworded it to "trend change after 1990" throughout the manuscript. We disagree, however, that the issue is simply an "instrument error". To our understanding the measurement sampling protocol was changed at the time, with 6 instead of 3 measurements per day. Difficult for further analysis is that the new homogenized data from Yang et al., 2018 have so far not been included in the GEBA database. The revised section 4.1 explains now to our best knowledge what we know and not know. We believe the gap filled GEBA data set we use can not to be discarded yet. As the reviewer rightfully points out, we do see a "jump" also in Japan, albeit delayed wrt to China. We have no indication that Japan went through a nation wide instrument upgrade or change in measurement

protocol at this point. Noticeably Norris and Wild (2009) pointed to possibly, different cloud cover trends in China and Japan making the interpretation of the respective SDSR trends and timings difficult. Figure 1b in Wang and Wild (2016) also shows that multiple sun duration measurements recorded the chinese "jump" in addition to the SDSR instruments. We do not have access to the data used in Yang et al. (2018), and we would need both access and further evaluation of both the GEBA data and the homogenized data to favor one over the other. We have therefore chosen to reduce the focus of the trend reversal, respectively jump, without fully excluding it as a possibility. What remains is a signifcant trend change around the year 1990. We do, however, agree with the reviewer that the trend reversal was still given too much emphasis in the revised version of the manuscript, so we have reduced the emphasis even further.

Regarding the emissions trend of SO2 (Figure A2 in the newer manuscript) versus the trend in the homogenized data (Green line in Figure 10 in Yang et al. (2018)) there are still apparent differences between the two. Emissions of SO2 show a flattening in the end of the 1990s just before a very strong increase until again flattening in 2005, while even the new homogenized chinese SDSR data show arguably no strong trend changes between 1995 and 2010. We therefore conclude in the newer manuscript:

*The modeled emissions of $SO_2$ as shown in Figure A2 over China showed no trace of a significant change in trend after 1990 in our observed SDSR timeseries as discussed in the previous section.*

**Reviewer Point P 1.2** — However, given the above I do not believe that this finding would change the main outcome of this paper much in that CMIP6 models underestimate the observed dimming signal over Asia and are unable to reproduce the recent brightening potentially due to uncertainties in the emission inventories. However, I think the direction of and timing of the temporal trends in observed and simulated SDSR over China (and Asia) would possibility match up better than with those observations currently presented in the manuscript i.e. the new observations show a dimming signal until 2005 and then brightening. I was wondering if there is anyway to include the dataset from Yang et al., (2018) on any of the figures in the manuscript to provide a more direct comparison of the new and old observations with the model simulations? Also, I think some minor changes to text to include more reference to this observational change would be beneficial. This could go some way to addressing the current comparison for China (and also for Asia in Fig 2c), where the obvious discrepancy in the figures is the sharp change in observations in the 1990s, which we now know was not really observed.

**Reply**: Unfortunately the dataset from Yang et al.(2018) is not available to us, and we refer to our reply to point P1.1 in that we do not disregard the observations of the "jump" completely. Without further studies evaluating the two observational datasets we reach the same conclusion as the reviewer in that models do not accurately represent observations, independent of which observational dataset we choose to compare with. Section 4.1 includes a brief update of our analysis based on the homogenized data:

*We can use Figure 2(f) to compare our model data to these homogenized data, and see that even without a larger "jump" in the data there are still large discrepancies between model and observation, both in the shape and magnitude of the brightening period after 1990. All models show dimming in the flattening period of the new homogenized data. All models apart from CanESM5 show an averaged negative trend between the 6-year-means of 2003-2008 and 2009-2014, where the homogenized data show a brightening. Models do not accurately represent the strength of dimming, or the time evolution*

*of SDSR observed in China with or without the early 1990s brightening.*
We underline that further studies on the observational datasets would prove very beneficial for the studying of model performance such as in our study.

**Reviewer Point P 1.3** — As mentioned above the CMIP6 models would still underestimate the magnitude of the dimming signal across China and do not represent the observed brightening from 2005, highlighting that there are uncertainties in the magnitude and timing of emissions over China. Therefore, potentially including some brief discussion on the recent literature of emission estimates of SO2 over China, particular for recent years, would prove beneficial and aid the conclusions of the paper regarding SO2, SO4 and SDSR (e.g. Lu et al., 2010; Koukouli et al., 2018; Zheng et al., 2018).

**Reply**: We thank the reviewer for these excellent paper suggestions. Koukouli et al. (2018) used satellite observations to make a new estimates for $SO_2$ emission between 2005 and 2015, and Lu et al. (2010) used technology based methodology to make estimates from 2000 to 2008. Figure 1 attached to to these author replies show the CMIP6 emission of $SO_2$ in China as diagnosed by four of the models in the study, together with the estimations found in Koukouli et al. (2018) and Lu et al. (2010).
These results has been taken into account and a brief discussion has been added in the newest version of the manuscript:
*Previous studies estimating $SO_2$ emissions include Lu et al. (2010), that found sulfur dioxide emissions in China increased by 53 % between 2000 and 2006 using technology based methodology, and thereby found similar results to that of Hoesly et al. (2018). Lu et al. (2010) also compared AOD derived SDSR to GEBA based SDSR data as shown in streets et al and found the GEBA based SDSR data to not accurately represent SDSR development in East Asia, this further underlines the need for more studies evaluating SDSR observations. Other studies such as Koukouli et al. (2018) have used satellite observations to estimate a new emission inventory for SO2 between 2005 and 2015 in China. We note that the year 2005 in Figure A2 is directly after the sharp increase in $SO_2$ emissions, and the biggest difference between the estimation made by Koukouli et al. (2018) and the $SO_2$ emission inventories in CMIP6 are a decrease in emissions after the year of 2011. This decrease in $SO_2$ emissions would intuitively result in a brightening, which is identified over the same time period in the homogenized data by Yang et al. (2018)(Fig 10 therein).*

**Reviewer Point P 1.4** — A general comment (and mentioned specifically below) is to consider making more reference in the results to the fact that aerosols influence multi-decadal trends in SDSR and other variables (clouds) are more important for short term variability.

**Reply**: This has been added as is mentioned in the replies to the minor comments below.

**Minor comments**

**Reviewer Point P 1.5** — Within the revised manuscript there appear to be references to Figures, tables and Appendix items that need to be updated. For example, the results section refers to a Figure 5 but I have not seen a Figure 5 in the revised manuscript. Also there does not appear to be a reference to Figure A1 in the text.

**Reply**: These reference errors are not present in the actual new revised manuscript, but are present only in the "track changes" file that was added in the author response. The "track changes" file was made using the latex function `diff` which do not always handle references between two manuscript versions correct, as has been made clear here by the reviewer. We doubled checked references and they appear to be correct now.

**Reviewer Point P 1.6** — Page 2 line 31-33 – Not sure I understood this sentence properly, could do with clarifying a bit

**Reply**: We have tried to clarified the sentence:
*Extraterrestrial influences like the 11-year cycle of the sun have not created any important trends on decadal time scales in Earths surface solar radiation in the past century (Eddy et al., 1982; Wild, 2009).*

**Reviewer Point P 1.7** — Page 2 lines 39-41 – Could link to the updated IPCC definitions, aerosol radiation interactions (Ari) and aerosol-cloud interactions (Aci).

**Reply**: Both terms ari and aci has been added with a reference to IPCC

**Reviewer Point P 1.8** — Page 5 line 140 – I think hist-piNTCF does not fix CH4 at pre-industrial levels so this might need to be removed from the description.

**Reply**: According to Table 1 in Collins et al. (2017) hist-piNTCF has historical development of CH4, so the reviewer is right and this has been fixed in the manuscript.

**Reviewer Point P 1.9** — Page 7, line 213 – remove 'the' before 1961

**Reply**: Fixed.

**Reviewer Point P 1.10** — Page 8 line 229 – remove 'between'

**Reply**: Fixed.

**Reviewer Point P 1.11** — Page 9 line 258 – perhaps refer to multidecadal dimming signals

**Reply**: Thank you for the suggestion. The sentence has been changed to:
*SDSR in the experiments hist-nat and hist-GHG do not show signs of dimming or brightening over the investigated period in China, which confirms that water vapor or stratospheric aerosols are not the dominant cause for multidecadal dimming signals in the fully coupled historical model simulations. This is supported by previous work as mentioned in the introduction.*

**Reviewer Point P 1.12** — Page 9 line 261 – perhaps include 'changes in' before anthropogenic aerosol emissions'

**Reply**: Added.

**Reviewer Point P 1.13** — Page 9, line 274-275 – perhaps again include reference to 'multidecadal trends in' all sky SDSR

**Reply**: Added.

**Reviewer Point P 1.14** — Page 9, line 283-284 – but could this be due to the errors in the observations pointed out in Section 4.1?

**Reply**: See our revised discussion in the new Section 4.1 where we do not disregard the original GEBA data (reply to P1.1.) but de-emphasize the reversal term.

**Reviewer Point P 1.15** — Page 10 line 310-311 – Potentially remove/reword 'not by changing cloud cover' as could be misleading. Yes aerosols change the brightness of clouds but what about aerosol the effect on cloud lifetime?

**Reply**: We agree this sentence was misleading and has been rewritten to:
*The aerosol indirect effect changes the radiative properties of clouds in two ways, by making them appear brighter, and by altering their lifetime (Boucher et al., 2013). Therefore, a weak change in cloud cover followed by a strong change in all sky and clear sky SDSR points to both the direct and the brightening indirect aerosol effect being the primary cause of SDSR change, as an altered lifetime of clouds would imply cloud cover changes.*

**Reviewer Point P 1.16** — Page 11 lines 329 – The calculation of the impact of changes in cloud cover on SDSR is presented but not really discussed. A simple sentence on the implications of this finding would improve its usefulness.

**Reply**: This calculation was made to give the reader an idea of the order of magnitude of how much cloud cover changes can affect SDSR, but can not be used to correct the SDSR model results for the effect of cloud cover changes as we do report complex simulations and observations in Table 2. We did add a sentence explaining the implications of this finding, as the reviewer suggested.
*A rough calculation of the effect of 1 % cloud cover increase on SDSR in China is found in Section A3 in the Appendix, indicating that such an increase could result in a dimming of 1-3.5 $Wm^{-2}$. As such it shows that observed and modelled changes in cloud cover, as reported in Table 2, can lead to important contributions to the dimming and brightening signals in SDSR. However, this calculation is idealized, does not isolate the cloud cover change effect in the model results and does not explain the inconclusive data reported in Table 2.*

**Reviewer Point P 1.17** — Page 11 line 336-337 – Based on the all-sky and clear-sky results can you say then that the dimming is primarily caused by direct aerosol radiation interacts, with a smaller impact from aerosol-cloud interactions?

**Reply**: If one has an all sky SDSR and clear sky SDSR anomaly of equal magnitude in a cloud free atmosphere, we could say the direct effect (ari) was dominating the all sky SDSR anomaly. However, if the atmosphere is cloudy, one cannot draw the conclusion that the direct effect (ari) was dominating over aci. All sky SDSR is not the sum of clear sky SDSR and cloud SDSR anomalies, as aci may reflect beams of radiation that would have been reflected by ari given clear sky conditions. Therefore it is impossible to say whether aci or ari dominates all sky SDSR anomaly in China, when we know the country is not cloud free. This is why we write "aerosol effects" are dominating, including both ari and aci.

**Reviewer Point P 1.18** — Page 11 line 340 – should you not specifically mention sulphate aerosols here, as not all aerosol scatter shortwave radiation?

**Reply**: This is the introductory sentence to the sulfate section, so we added the word "reflective" so clarify that we are not talking about all aerosols.
*In the atmosphere, the presence of a reflective aerosol is the cause for scattered shortwave radiation, and the emission of its precursor is only an indirect indicator of its presence.*

**Reviewer Point P 1.19** — Page 12 line 358-359 – Based on section 4.1 can we say that GEBA, as it is presented in Figure 4b provides a reasonable representation of the historical development of SDSR over China?

**Reply**: See our revised discussion in the new Section 4.1 where we do not disregard the original GEBA data (reply to P1.1.) but de-emphasize the reversal term. We have modified the sentence slightly:
*Assuming GEBA data provide a reasonable representation - within uncertainty bounds as discussed in section 4.1 - of the historical development of SDSR...*

**Reviewer Point P 1.20** — Page 15 line 458 – should we expect a trend reversal in emissions given that this was mostly identified as an observation error in section 4.1?

**Reply**: As discussed above we de-emphasize the term reversal, but we do not disregard the original GEBA data. We point out that the modelled $SO_2$ emissions do not fit with neither homogenized data (Yang et al., 2018) nor gap filled GEBA. Sentence has been simplified to:
*The modeled emissions of $SO_2$ as shown in Figure A2 over China showed no trace of a significant change in trend after 1990 in our observed SDSR timeseries as discussed in the previous section.*

**Reviewer Point P 1.21** — Appendix A3 Line 518 – makes reference to rsds twice

**Reply**: One of them was meant to say "rsdscs", this has been fixed.

[Figure]

Figure 1: Emission of $SO_2$ in China, diagnosed by four of the models in this study. China is defined here as the area within latitudes [20°N–45°N], and longitudes [95°E–125°E]. In addition the results from Koukouli et al. (2018) and Lu et al. (2010) has been added.

[revised manuscript text omitted]